# MagmaFOAM-1.0: a modular framework for the simulation of magmatic systems

Federico Brogi[1,2], Simone Colucci[1], Jacopo Matrone[3], Chiara P. Montagna[1], Mattia De' Michieli Vitturi[1,4], and Paolo Papale[1]

[1]Istituto Nazionale di Geofisica e Vulcanologia, sezione di Pisa
[2]Istituto Nazionale di Oceanografia e di Geofisica Sperimentale
[3]Dipartimento di Matematica, Università degli Studi di Firenze
[4]University at Buffalo, Department of Geology

**Correspondence:** F. Brogi (federico.brogi@ingv.it)

**Abstract.** Numerical simulations of volcanic processes play a fundamental role in understanding the dynamics of magma storage, ascent and eruption. The recent extraordinary progress in computer performance and improvements in numerical modeling techniques allow simulating multiphase systems in mechanical and thermodynamical disequilibrium. Nonetheless, the growing complexity of these simulations requires the development of flexible computational tools that can easily switch between sub-models and solution techniques. In this work we present MagmaFOAM, a library based on the open source computational fluid dynamics software OpenFOAM, that incorporates models for solving the dynamics of multiphase, multicomponent magmatic systems. Retaining the modular structure of OpenFOAM, MagmaFOAM allows run-time selection of the solution technique depending on the physics of the specific process, and sets a solid framework for in-house and community model development, testing and comparison. MagmaFOAM models thermo-mechanical non-equilibrium phase coupling and phase change, and implements state-of-the-art multiple volatile saturation models and constitutive equations with composition-dependent and space-time local computation of thermodynamic and transport properties. Code testing is performed using different multiphase modeling approches for processes relevant to magmatic systems: Rayleigh-Taylor instability, for buyoancy-driven magmatic processes; multiphase shock tube simulations, propedeutical to conduit dynamics studies; bubble growth and breakage in basaltic melts. Benchmark simulations illustrate the capabilities and potential of MagmaFOAM to account for the variety of non-linear physical and thermodynamical processes characterizing the dynamics of volcanic systems.

## 1 Introduction

Simulating transport processes in volcanic systems is of crucial importance to understand the physics of eruptions, correctly interpret geophysical signals recorded by volcano monitoring systems, anticipate volcanic scenarios, and forecast volcanic hazards (Sparks, 2003; Bagagli et al., 2017). A great number of flow models have been developed to address specific volcanic processes, including magma chamber dynamics (Ruprecht et al., 2008; Bergantz et al., 2015; Garg et al., 2019), conduit flow (Melnik, 2000; Papale, 2001; de' Michieli Vitturi et al., 2008b; Colucci et al., 2017b), volcanic plumes (Suzuki et al., 2005; Cerminara et al., 2016), pyroclastic flows (Esposti Ongaro et al., 2007; de' Michieli Vitturi et al., 2015; Dufek, 2016) and

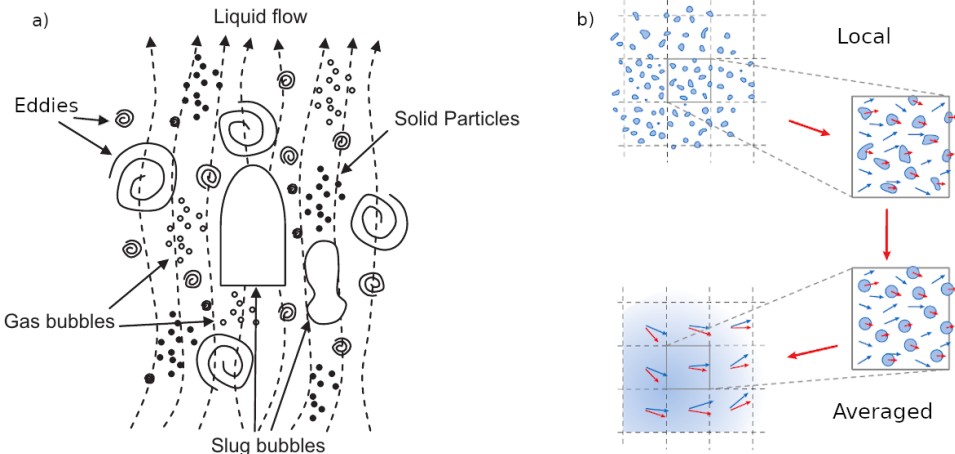

**Figure 1.** Schematic representation of: (a) multiscale nature of a gas-liquid-solid flow (reprinted with minor modifications from Yeoh and Tu (2019b) with permission from Elsevier); (b) a two phase volume at the local scale containing discrete phase costituents (e.g. bubbles or droplets) and the corresponding approximation path to describe it as made of two interpentrating continuum fluids (modified from Marschall (2011))

lava flows (Griffiths, 2000). Inter-model comparison studies have evaluated individual model performance and the relevance of the different subprocesses, and have highlighted target areas for improvement (Massol and Koyaguchi, 2005; Macedonio et al., 2005; Sahagian and Proussevitch, 2005; Costa et al., 2016). All these models attempt to tackle the great complexity arising from the presence of multiple phases. Interactions among liquid phases (e.g. silicate melt), solid phases (e.g. crystals or pyroclasts) and gas phases (exsolved volatiles or atmospheric gas) are indeed ubiquitous in volcanic systems, from deep magma chambers up into the atmosphere (e.g., Jackson et al., 2018; Keller and Suckale, 2019).

Volcanic transport processes are typically characterized by a wide range of spatial and temporal scales at which different interacting physical subprocesses occur (Griffiths, 2000; Gonnermann and Manga, 2007; Dufek, 2016). From a modeling perspective, there is no general approach able to treat all these subprocesses at the same time, thus specific models are usually developed for each application.

A generic multiphase system can be thought of as composed by sub-domains or regions pertaining to single phases (gas, liquid or solid), separated by interfaces (boundaries) representing sharp discontinuities where the physical properties change abruptly. The main challenge in modeling multiphase with respect to single phase flows is due to the presence of such a discontinuity. The topology of this interface defines the amount of interfacial area that is available for the phases to exchange mass, momentum and energy and strongly affects the behavior of the multiphase mixture. Moreover, this interface is not static but changes dynamically with the flow and complex flow features may emerge due to the presence of moving phase boundaries. Understanding and modelling multiphase flows also requires taking in appropriate consideration their multiscale character. The typical size of the interfaces can be comparable to, or orders of magnitude smaller, than the domain and flow length scales; or even cover a broad range of scales (Figure 1). Cascading effects and multiple coupled phenomena at the different scales may

have dramatic consequences on the flow dynamics. At the scale of the system (e.g. volcanic conduit), large flow structures that govern the flow directly depend on the properties of the multiphase mixture, that in turn are determined by the dynamic reorganization of local mesoscale structures (e.g colascense/breakage of large bubbles or bubble cluster dynamics) as well

as the motion of individual constituents at the microscale (e.g. single small bubbles) within a continuum phase (e.g. liquid). Modeling such complex phase interactions across a wide range of scales represents certainly a big challenge.

Interface-resolving methods, similar to direct numerical simulation (DNS) approaches in single-phase turbulent flows (Moin and Mahesh, 1998), fully resolve the scales of the fluid equations and track the topology of the interfaces. With this approach no assumptions are made regarding the properties of the multiphase mixture or interfacial phase exchanges. The dynamics of

the multiphase system emerges naturally from the computation as a direct consequence of solving phase interactions locally at the interfacial scale (under the constraints of mass, momentum and energy conservation for each phase). DNS therefore can be thought of as a virtual laboratory to understand fundamental physics, especially at the micro scale, that can capture emerging dynamics resulting from non linear phase interactions that are difficult to be parametrized a priori (e.g. Segre et al., 2001). DNS can also provide a detailed description of the flow that often is not accessible in experiments. Therefore it can help in

interpreting laboratory observations (e.g. Qin and Suckale, 2020) or even building and testing constitutive/parametric models for interphase interaction exchanges and the behaviour of the multiphase mixture as a whole (Fang et al., 2019). In volcanology, the DNS approach has been used to study large gas bubbles ascending in a conduit through low viscosity melts (Suckale et al., 2010a), buoyancy-driven instabilities in liquids at different densities (Suckale et al., 2010b), as well as the complex rheological behaviour of crystal bearing magmas (Qin and Suckale, 2020). Based on the computationally efficient Lattice Boltzmann

method, interface resolving modeling has been also useful to better understand bubble growth, deformation and coalescence (Huber et al., 2014) as well as the mush microphysics characterizing crystal-rich magma reservoirs (Parmigiani et al., 2014). Despite its proven ability to provide understanding of the fundamental physics near interfaces, DNS remains limited to specific, computationally tractable problems, since it requires a large amount of computational resources.

Simulations of magmatic systems that can aid the interpretation of geophysical (Bagagli et al., 2017) or petrological observa-

tions (Cheng et al., 2020), need to cope with very large domains, on the scale of the kilometers. On the other hand, the smallest scales (e.g small eddies, bubbles, crystals, Figure 1a) remain several orders of magnitudes smaller, resulting in skyrocketing computational costs for DNS even when considering relatively simple flow conditions (e.g. laminar flows, Yeoh and Tu (2019a)). Multiphase flows that present dispersed interfaces (e.g. bubbly, droplet or particle-laden flows, Figure 1b), common in both natural and industrial settings (Keller and Suckale, 2019; Moreno Soto et al., 2019), have been successfully modeled

with a different approach. The multi fluid formulation employs averaging techniques that filter out the interfacial scales that are too small to be resolved (Marschall, 2011). The complexity of a volume with multiple phases at the local scale is characterised by phase-average properties and a volumetric fraction that expresses the relative presence of one phase with respect to the others (Figure 1b). Neglecting the details of the topology of the interfaces at the local scale allows to describe the phases at the system scale as interpenetrating continua governed by separate sets of conservation equations. The resulting equations

hence resemble those for single phase flows except for the volumetric fraction and the presence of phase interaction terms that require appropriate closure. Similarly to Large Eddy Simulations for turbulent flows, additional constitutive models are in fact

required to recover the physics of the missing small scales. The multi-fluid approach allows modeling thermo-mechanical dis-equilibrium (e.g. phases with different velocities, temperatures or pressures) as well as interactions of the dispersed phases for any multi-phase system (Marchisio and Fox, 2007), including magmas (Keller and Suckale, 2019). Applications of multi-fluid modeling in volcanology include but are not limited to the study of buoyancy-driven magma mixing (Ruprecht et al., 2008), conduit dynamics (Papale, 2001; Dufek and Bergantz, 2005) and volcanic plumes (Neri et al., 2003; Ongaro et al., 2007). However, the definition of appropriate closure models for interfacial phase exchanges is crucial for these models to provide accurate predictions on the evolution of the multiphase system. The closure of the multi fluid formulation remains challenging and is mostly achieved using system-dependent constitutive equations, often empirical, valid for specific flow regimes (in terms of interface topology and/or concentrations of the dispersed phase). Thus the generality of the multi fluid formulation is reduced by the specificity of the constitutive models. Multi fluid models are, however, more computationally expensive than single phase models, as they require an additional set of governing equations for each phase. As the number of phases increases, the computational burden also increases dramatically (e.g. Ferry and Balachandar, 2001). The definition of the interfacial exchange terms can also indirectly increase the computational cost. For instance, the fluid-particle drag introduces a time scale in the equations, the relaxation time of the dispersed phase, that describes the time required by the particle to adapt to a change in velocity of the surrounding fluid. When this relaxation time is small (typically for small particles and/or high fluid viscosities), the stability and accuracy of the numerical solution require a time step smaller than the relaxation time, increasing the number of iterations needed to solve the flow time scale. Under the assumption of thermo-mechanical equilibrium the equations of the multi fluid model can be further reduced to an evolution equation for a single pseudo-fluid representing a mixture of multiple phases. From a computational point of view, given the reduced number of equations needed to track the evolution of the mixture, this is a more convenient approach. In addition, when there is a strong thermo-mechanical coupling between phases (small relaxation times), it is reasonable to assume that the particle velocity is equal to the fluid velocity, effectively removing the aforementioned issues related to the definition of the interaction terms and the relaxation time. The single mixture, pseudo-fluid approach has received some success within the volcanological community. Simulations of magma mixing (Longo et al., 2012; Garg et al., 2019) and conduit dynamics (de' Michieli Vitturi et al., 2008a; Melnik and Sparks, 2006) as well as volcanic plumes (e.g., Suzuki et al., 2005) are only a few examples of application. Nevertheless, as for the multi fluid, constitutive models play a crucial role in determining the reliability of single fluid predictions. Finally, it is also important to emphasise that both the multi fluid and the single fluid mixture models are based on the average form of the multiphase flow equations. Averaging procedures implicitly require the separation of scales. In volume averaging for instance, the volume should be large enough to contain a representative sample of the dispersed phase (e.g. bubbles or particles) but much smaller than the typical distance over which flow properties vary significantly. This condition is rarely satisfied in real application since intermediate scales between the local and the system scale are present (Brennen and Brennen, 2005). In spite of all the issues and limitations, among which few have been mentioned above, a large amount of theoretical and practical investigations remain based on single and multi fluid models (Yeoh and Tu, 2019a).

The increased ability of models to include detailed physics strictly requires the development of more flexible computational tools that can easily switch between constitutive models and solution techniques to adapt to different dynamical regimes,

thereby reducing computational efforts, increasing usability and easily allowing scientists to perform inter-model comparison studies and models coupling.

The open source library OpenFOAM provides a variety of fluid solvers for multiphase flows, that can be combined with several different constitutive equations. Its modular object-oriented implementation allows the developers to easily expand and adapt the code, and the users to combine different models at run-time with almost no need to code. Given a set of discretised fluid evolution equations (or 'solver'), the user can easily select appropriate thermophysical and rheological models or switch from 2D/3D to axis/plane symmetric simulations. The OpenFOAM community is continuosly growing, as is the range of applications of interest for both the academy and industry (e.g., Winden, 2021). Moreover, the recently established exaFoam consortium will improve computational performance enabling the "OpenFOAM community to exploit efficently the current evolving HPC hardware and middleware" (www.exafoam.eu). OpenFOAM is thus an ideal platform for developing a computational toolbox for the next generation of magmatic systems modeling. In this work we present the MagmaFOAM library, an extension of OpenFOAM dedicated to solving multiphase volcanic flows. The current implementation features multiple volatile saturation models (Papale et al., 2006) and specific formulations for the equation of state (Lange and Carmichael, 1987) and viscosity (Giordano et al., 2008) of magmatic mixtures including dissolved volatiles. MagmaFOAM retains the basic coding principles of OpenFOAM, inherits its flexibility and takes full advantage of the family of fluid solvers and constitutive models (e.g. non-Newtonian rheological models) already implemented in OpenFOAM.

This paper is structured as follows. First we provide an overview of the basic ingredients of MagmaFOAM, including the specific magmatic constitutive equations and how they are implemented. Then, we show benchmarks and validation tests aimed at verifying the code ability to solve problems for segregated and dispersed flows of interest for magmatic systems with different modeling approaches. Finally, we summarize and discuss our results and draw the conclusions.

## 2 MagmaFOAM ingredients

### 2.1 Structure of MagmaFOAM

MagmaFOAM uses the same organization of OpenFOAM (Figure 2) and its hierarchy is therefore subdivided into applications and libraries (`src`). Code organization is therefore rational and efficient, reducing code duplication, promoting code reusage and facilitating testing. Most of the applications are assembled at run-time based on the user requests using dynamic linking to pre-compiled libraries: before running a simulation the user can arbitrarily select boundary conditions, discretization schemes, mixture and phase constitutive equations. This mechanism allows selecting and combining modeling ingredients, among the possible combinations, from both OpenFOAM and MagmaFOAM (Figure 2), without the need of coding.

### 2.1.1 Multi-component constitutive models for magmatic systems

The dynamics of magmas as they ascend, stall through the crust and possibly erupt is strongly dependent on their physical properties (mostly density $\rho$ and viscosity $\mu$), which in turn are determined by composition and phase distribution, pressure $p$

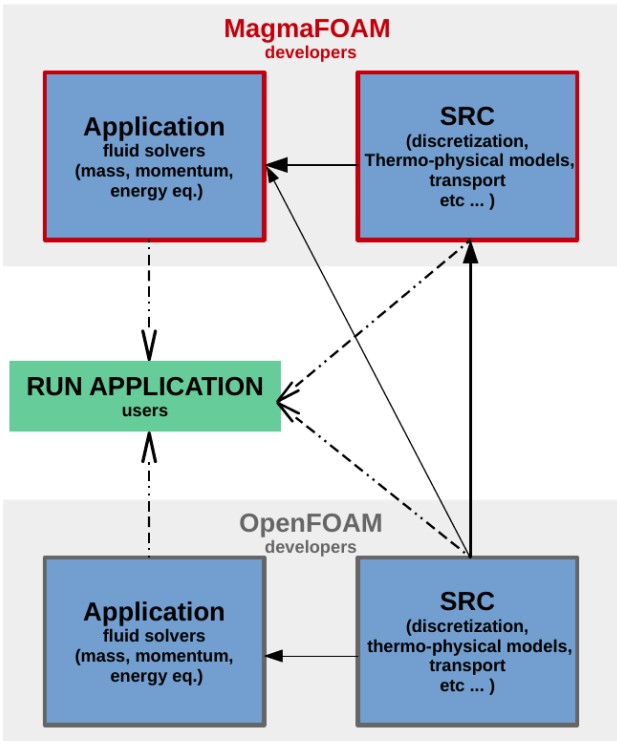

**Figure 2.** MagmaFOAM - OpenFOAM coupling scheme.

and temperature $T$ conditions. The interplay among $p$-$T$ conditions, melt-crystals-bubbles phase changes and density and viscosity variations originates a wealth of possible space and time patterns for magma storage, transport and eruption (e.g., Lesher and Spera, 2015). When handling this thermo-physical complexity, state-of-the-art multi-component constitutive models that compute melt properties as a function of the local pressure, temperature and composition are the necessary basic ingredients and have been implemented in MagmaFOAM.

Multi-component volatile saturation is included through the SOLWCAD model (Papale et al., 2006), which provides equilibrium $H_2O$-$CO_2$ saturation over a broad range of $p-T$ conditions and for virtually any melt composition. This model overcomes the ideal Henrian behaviour, which is a reasonable approximation only at low pressures ($\lesssim 100\,\mathrm{MPa}$; Papale et al., 2006). Once the phase distribution of volatile species is computed through SOLWCAD, the relevant physical properties for the multiphase magma can be derived.

The density of the silicate melt up to a few GPa ($\lesssim 3\,\mathrm{GPa}$ or $\lesssim 100\,\mathrm{km}$ depth) is computed as in Lange and Carmichael (1987) with an empirical equation of state, as a ratio of the oxides' molar masses ($M_i$) and molar volumes ($V_i$):

$$\rho(p,T,X) = \frac{M(X)}{V(p,T,X)} = \frac{\sum_i X_i M_i}{\sum_i X_i V_i(p,T)}, \tag{1}$$

where $X_i$ is the mole fraction of the $i$-th oxide component. To a good approximation, molar volumes do not depend on melt composition (Lesher and Spera, 2015) and can be computed with a polynomial expansion:

$$V_i(p,T) = \sum_{l,m} a^i_{l,m} T^l p^m = a^i_{0,0} + a^i_{1,0} T + a^i_{0,1} p + a^i_{1,1} pT + ... \tag{2}$$

The polynomial coefficients $a^i_{l,m}$ have been determined from laboratory experiments. For the oxides we have used the coefficients reported by Lange and Carmichael (1987) and Lesher and Spera (2015). For $H_2O$ and $CO_2$ we referred to Burnham and Davis (1974) and Papale (1999), respectively.

Melt viscosity is described as in Giordano et al. (2008). This model includes temperature and compositional effects for a wide range of melt compositions. In addition, the model can be used to determine the compositional dependence of important viscosity-derived properties, such as melt glass transition temperature and fragility. This aspect may be particularly relevant when modeling the ascent of degassing magma to determine the potential for brittle fragmentation. A drawback is that the model does not take into account the effect of pressure on viscosity, which can become relevant when modeling magma transport in the deep crust and mantle.

The model is based on the Tammann-Vogel-Fulcher (TFV) relationship for the non-Arrhenian temperature dependence of the bubble-crystal free viscosity $\eta$:

$$\log \eta = A + \frac{B}{T - C}, \tag{3}$$

where $T$ [K] is the temperature, $A$ is a constant and $B$ and $C$ are parameters that depend on the melt composition, including dissolved volatile species. The $A$ constant provides a high temperature limit for viscosity ($\sim 10^{-4}\,\mathrm{Pa\,s}$), that holds for all melts regardless of their composition and is supported by both theoretical considerations and experimental observations (e.g. Scopigno et al., 2003). Let us also note that Equation 3 has no physical meaning for $T \leq C$ (Mauro et al., 2009).

## 2.2 Modeling volatiles concentration at the bubble-melt interface

Models accounting for multicomponent phase change require a description of the evolution of the composition at the interface between phases. The mass transfer rate (per unit volume of liquid+gas) of a volatile component can be defined as the product between the interfacial mass flux $J_i$ [kg/(m$^2$s)] and the interfacial area concentration $A$ [m$^2$/m$^3$]

$$\Gamma_i = J_i A \tag{4}$$

The area concentration $A$ is determined by the geometrical configuration of the gas-liquid interface and hence it is strongly dependent on the flow regime. It can be computed using simple geometrical assumptions on the dispersed phase (e.g. monodisperse bubbles with constant radius) or, for more complex flow scenarios, with additional transport equations (e.g. IATE model (Ishi and Hibiki, 2006)). The model for $J_i$ expresses the driving force for diffusive mass transfer of the component $i$ and can

be calculated with the following relationship

$$J_i = k_i \Delta C_i \tag{5}$$

where $k_i$ [kg/(m$^2$s)] is the mass transfer coefficient, a function of the diffusion coefficient $D_i$ (Cussler, 2009; Thummala, 2016), and $\Delta C_i$ is the difference between the mass fraction of the specie in the phase ($C_i$) and at the interface ($C_i^{\mathrm{f}}$):

$$\Delta C_i = C_i - C_i^{\mathrm{f}}. \tag{6}$$

Under the assumption of local equilibrium, the mass fraction at the interface can be expressed as

$$C_i^{\mathrm{f}} = C_i^{\mathrm{sat}}(p, T_{\mathrm{f}}, X, X_{Vtot}), \tag{7}$$

where $C_i^{\mathrm{sat}}$ is the saturation concentration of a specific volatile specie (i.e. mass fraction at thermodynamic equilibrium). In general this is a non linear function of pressure ($p$), temperature at the interface,($T_{\mathrm{f}}$), melt composition ($X$) and total amount of volatiles of all species ($X_{Vtot}$). For magmas with $H_2O$ and $CO_2$, $C_i^{\mathrm{sat}}$ can be computed using SOLWCAD (Papale et al., 2006) or other dedicated models (e.g., Newman and Lowenstern, 2002; Burgisser et al., 2015). Direct coupling of any fluid solver with these models is usually too computationally expensive. Therefore, MagmaFOAM solvers can read the saturation surface from a pre-processed table. During the simulation, tabulate values are interpolated (multilinear interpolation) and used to compute $C_i^{\mathrm{sat}}$ in Equation (7).

## 2.3  MagmaFOAM constitutive models

Constitutive models implemented in MagmaFOAM can be selected and combined at run-time (no need of coding) with existing OpenFOAM solvers suitable for the specific problem under consideration (Figure 2). For example, the MagmaFOAM model for silicate melt density can be used with any compressible solver, either single- or multi-phase. This constitutive model is not compatible with incompressible solvers, that require density to be constant; however, in this case the density of the incompressible fluid can be preliminarily defined taking advantage of the dedicated MagmaFOAM utility `magmaThermoMixture`. The latter can also be used for testing implemented models as it simply returns the thermophysical properties as a function of composition, pressure and temperature. Demonstrative tutorials are included in MagmaFOAM to show how the end user can accomplish all these tasks at run-time using both single-phase and multiphase solvers.

## 2.4  Models for multicomponent bubble growth

Volatiles' phase changes and bubble growth are ubiquitous processes in volcano dynamics (Proussevitch and Sahagian, 1998). The gas exsolution process begins with the nucleation of bubbles in an oversaturated melt and continues with bubble growth. Bubbles grow by mass diffusion, when the silicate melt is oversaturated in volatiles, and by mechanical expansion as a response to pressure decrease. The viscosity of the surrounding melt and the surface tension oppose a resistance to bubble growth and control the mechanical disequilibrium between the bubbles and the melt itself. A number of works (Proussevitch et al., 1993; Lyakhovsky et al., 1996; Proussevitch and Sahagian, 1998; Lensky et al., 2001, 2004; Chouet et al., 2006; Shimomura et al.,

2006; Coumans et al., 2020) solve the system of bubbles as a monodisperse periodic array of static, spherical, single-component

($H_2O$) growing bubbles surrounded by a viscous melt shell, using the Rayleigh-Plesset equation. A suite of models, based on a similar approach, have been implemented in MagmaFOAM and benchmarked to simulate multicomponent diffusive bubble growth. This approach provides, at low computational cost, an accurate representation of the coupled momentum balance and diffusive transport of volatiles, because it well resolves the concentration profile near the bubble interface (Huber et al., 2014). The strong assumptions that the size distribution is monodisperse and the bubbles are non deformable and mechanically

coupled with melt, limits the range of applicability of the model. In high-viscosity systems at low vesicularity, the model can provide reliable results when compared with experiments (e.g., Coumans et al., 2020). The model does not take into account interfacial interactions (fluid-particle and particle-particle) that can give rise to emergent behaviour, as in the case for example of bubble waves (Manga, 1996). All model equations can be found in Appendix B and are solved as a systems of ordinary differential equations (ODEs) using the OpenFOAM ODE solvers.

## 225  3    Benchmarks and test cases

The test cases presented here are included in the MagmaFOAM distribution together with the relevant post-processing routines. The results shown here are thus fully reproducible, and the benchmarks can be used to study the accuracy and efficiency of other OpenFOAM or external solvers.

### 3.1    Interface resolving modeling

The Volume of Fluid method (VOF) is adopted in OpenFOAM to resolve the position and shape of the interface separating two fluids or phases (e.g. liquid-gas). This methodology treats the interface discontinuity as a smooth but rapid variation (few computational cells) of an indicator field (volumetric fraction) representing the relative presence of one phase with respect to the other in each cell. The volumetric fraction is zero or one away from the interface, allowing to distinguish between one phase and the other, and assumes intermediate values in the region containing the interface. As a result, the location of the interface

and its shape are known only implicitly from the volumetric fraction. The evolution of the interface is then obtained by simply advecting the volumetric fraction using the velocity field computed from a single (e.g. the OpenFOAM solver `interFoam`) or multi-fluid momentum equation (e.g. the OpenFOAM solver `multiphaseEulerFoam`).The transport equation for the indicator function is under the constraint of mass conservation and therefore, with respect to other methods (e.g. Level-set method), VOF is mass conservative by construction. However, in practice the conservation of mass depends on the accuracy in

solving numerically the transport equation. The discontinuous nature of the volumetric fraction (a step function) at the interface makes the numerical solution of this equation challenging. In particular, numerical diffusion due to the discretization of the advection term prevents a sharp representation of the interface that tends to be smeared over the computational cells causing inaccurate estimations of its position and curvature. Different techniques exist to solve this issue. With a geometrical approach one may reconstruct the position of the discontinuity at the subgrid level, provided that the interface can be described with a

specific functional form (Rider and Kothe, 1998; Aulisa et al., 2003). The interface is then advected by the flow in a lagrangian

manner. This technique effectively prevents numerical diffusion and provides a more accurate representation of the interface at the cost of a significantly more complex algorithm and increased computational load. Other approaches rely on relatively more simple algebraic solutions that reduce numerical diffusion (e.g. Ubbink and Issa, 1999). Specifically, `interFoam` makes use of a high order differencing scheme (in the interface region only) and an additional compressive term in the advection equation that effectively counterbalances the numerical diffusion of the interface. While this approach is simpler and less computationally expensive than the geometrical reconstruction, the interface is spread over few computational cells and its precise position remains unknown. Nevertheless, in kinematic tests, interFoam has shown good mass conservation properties and acceptable advection errors (Deshpande et al., 2012). Spurious currents and artificial deformations of the interface are also an issue with VOF. Inaccurate interface curvature, together with a discrete force imbalance at the interface, typically produce spurious vortices that can artificially deform the interface. Depending on the simulation setup, the kinetic energy of these vortices may rapidly decay or grow and in the worst case scenario even cause the simulation to crash. However, spurious currents may pose a serious issue mostly for surface tension dominated flows and are less important for inertia dominated flows. For `interFoam`, (Deshpande et al., 2012) have shown that the growth of spurious currents can be controlled by choosing an appropriate time step. interFoam solves flows characterized by constant, or slowly-varying with respect to the flow time scales, fluid properties. This approximation holds for relevant volcanic scenarios as gas-poor magmatic reservoirs at depth, which are characterized by relatively fast overturn times (Ruprecht et al., 2008; Perugini et al., 2010; Montagna et al., 2015).

Here we present benchmarks and test cases to evaluate the accuracy of the solver `interFoam` to explore the dynamics of two immiscible fluids separated by a free interface. Specifically, we perform detailed studies of buoyancy-driven magma mixing and rising bubble dynamics. Overall, we find a remarkably good agreement between our simulation results and theoretical or numerical results from literature, over different flow regimes of interest for magma dynamics. The numerical solutions relative to cases with low Reynolds number Re are very accurate (e.g. Figure 4 and 7). At larger Re, the results are less accurate due to the appearence of high frequency numerical noise that can trigger secondary spurious interface instabilities (e.g. Figure 5. Reducing numerical noise by adopting different numerical schemes is one relevant element for future investigation. The magnitude of the compressive term, used in the solver to prevent numerical smearing of the interface, is a free parameter in the simulations and may influence the accuracy of the solution depending on the problem parameters. More recent OpenFOAM versions include more rigorous and accurate interface-resolving methods (e.g. Roenby et al., 2017).

### 3.1.1 Interacting magmas

Magma is thought to rise from the mantle into the crust in discrete batches (Annen et al., 2006) that then tend to stall and cool at different depths, while their chemistry evolves towards more felsic compositions (Sigurdsson et al., 2015). Different batches of magma may interact as they ascend towards shallower depths, resulting in magma mingling and mixing. The latter are widespread phenomena in volcanic plumbing systems (Perugini and Poli, 2012; Morgavi et al., 2017) and have often been invoked as eruption triggers (Wark et al., 2007; Druitt et al., 2012; Martí et al., 2020). Mingling and mixing are typically driven either by gravitational Rayleigh-Taylor instabilities, involving contacts between magmas with different densities due to compositional, thermal or phase stratifications (e.g., Jellinek et al., 1999; Montagna et al., 2015; Garg et al., 2019); or by

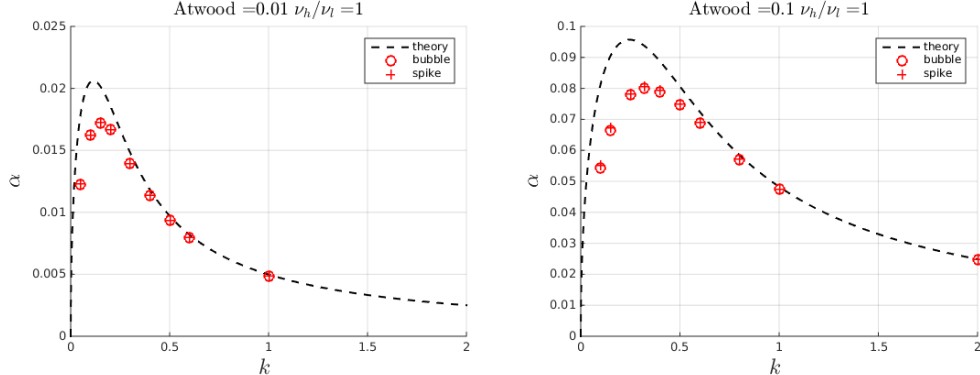

**Figure 3.** Comparison between computed growth rates (symbols) of the Rayleigh Taylor instability in the linear regime obtained with the solver `interFoam` and theoretical ones (dashed line). Bubble growth rates are computed tracking the position of the interface with respect to the central axis of the domain while spike growth rates are computed with respect to one of the lateral boundaries.

percolation of pressurized magmas arriving from depth into mushy reservoirs (Bachmann and Berganz, 2003; Seropian et al., 2018).

A standard benchmark to test numerical solvers for Rayleigh-Taylor instability problems requires to compare computed growth rates for small-amplitude single-mode perturbations with the linear stability theory. The latter predicts that a small perturbation grows exponentially with a rate that depends on its wavelength and on fluid density and viscosity contrasts (Chan-

285 drasekhar, 1955), surface tension (Chandrasekhar, 2013), compressibility (Mitchner and Landshoff, 1964) and diffusivity (Duff et al., 1962; Xie et al., 2017). The problem parameters can be expressed by two dimensionless numbers: the Atwood number $\text{Atw} = (\rho_h - \rho_l)/(\rho_h + \rho_l)$ and the Reynolds number ($\text{Re} = \sqrt{Wg}W/\nu$), where $\rho_h$ and $\rho_l$ are the two liquid densities, $\nu$ is the kinematic viscosity ($\nu_h = \nu_l$), $W$ is the wavelength of the perturbation and $g$ is the gravitational acceleration. We consider a 2D rectangular domain with a no-slip condition (walls) on top and bottom boundaries and periodic conditions on the sides. The

290 interface between the two liquids is located at the center of the computational domain (Figure 4). The size $L$ of the computational box is determined by the wavelength of the initial perturbation ($L = W \times 2W$). Benchmark results are reported in Figure 3 for Atwood numbers relevant for natural melts. The computed growth rates are in agreement with the theory (Xie et al., 2017) for different wavelengths (or equivalently wave numbers $k = 2\pi/W$) of the perturbation. The solver underestimates the peak growth rates at low $k$, corresponding to high $\text{Re}$. A more in-depth analysis of the results (Appendix A) reveals that this

discrepancy is mainly due to an initial delay in the onset of the perturbation. Removing this initial offset, the computed growth rates result much more accurate. Smaller initial perturbation amplitudes also improve accuracy.

As the instability grows and its amplitude becomes comparable with its wavelength, non-linear effects become dominant and the linear theory is not valid to predict the evolution of the system anymore. In order to validate `interFoam` for non-linear regimes we have compared our results with He et al. (1999) for single-mode perturbation with a 10% amplitude-to-wavelength

ratio, Atwood number $A = 0.5$ and Reynolds number $\text{Re} = 256$. A remarkably good agreement is obtained for the evolution of

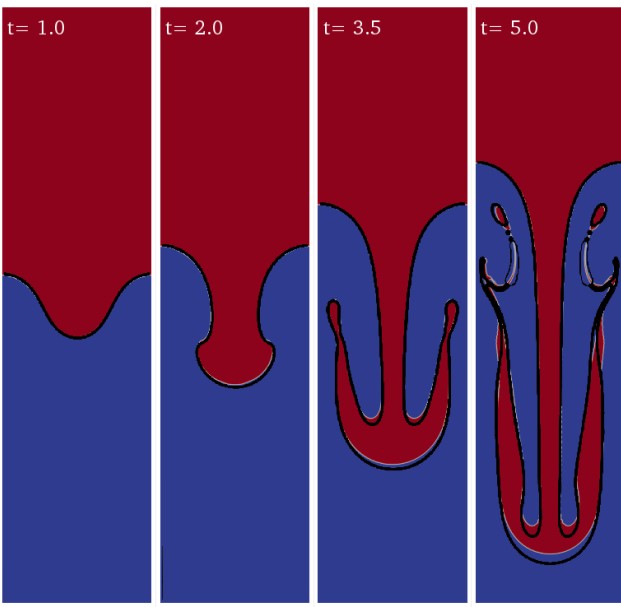

**Figure 4.** Rayleigh-Taylor instability (A = 0.5, Re = 256) computed with OpenFOAM solver `interFoam`. The density field (color-coded) is compared with the density countours in He et al. (1999) (black lines).

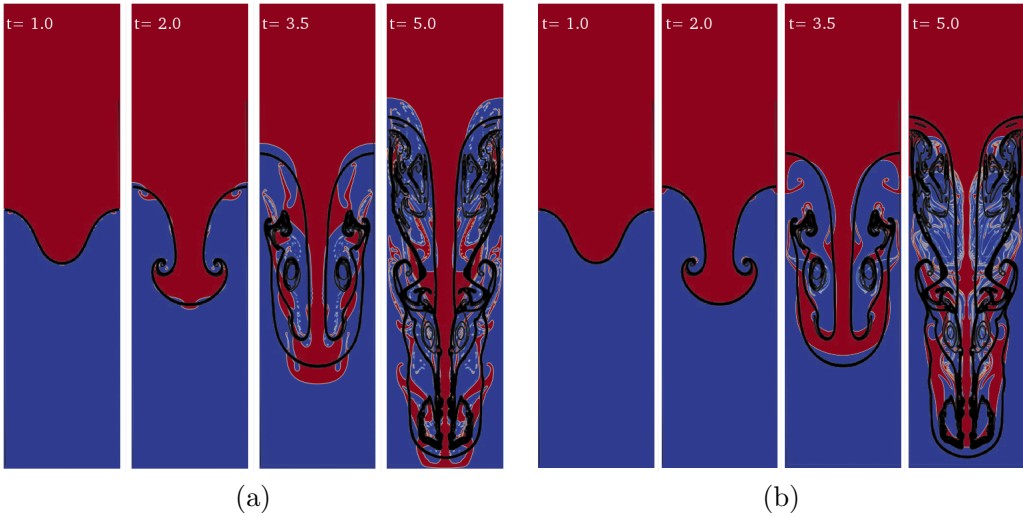

(a)                  (b)

**Figure 5.** Rayleigh-Taylor instability (A = 0.5, Re = 2048) computed with OpenFOAM solver `interFoam` (in color) with: (a) $256 \times 1024$ cells and default value for interface compression factor $C_\alpha = 1$; (b) $512 \times 2048$ cells and $C_\alpha = 0.1$. The density field (color-coded) is compared with the density countours of He et al. (1999) (black lines).

the fluid interface (Figure 4) using the same resolution (256 x 1024 cells). Convergence of the results was tested for different space and time resolutions using adaptive time-step based on maximum allowed Courant Number ($Co_{max} = 0.5$) to speed up the simulation. For a given mesh resolution the accuracy and convergence of the solution depend on the values of $Co_{max}$ and number of iterations (`nIter`) used to solve the pressure-velocity coupling with the PISO algorithm (Issa, 1986). Generally, larger $Co_{max}$ ($< 1$ for numerical stability) require larger `nIter` for solution convergence; our experience suggests that a relatively high number of `nIter` balances larger values for $Co_{max}$, reducing computational times. This way, even if the errors relative to the continuity equation are larger, the solution is not affected significantly.

For the high-Reynolds-number test case ($Re = 2048$) of He et al. (1999), the quality of the solution deteriorates using the same resolution (Figure 5). The interface is deformed by artificial secondary instabilities most probably triggered by spurious numerical noise. Removing the interface compression term and doubling the number of cells improves the solution to nearly match the reference.

Magmas usually interact both mechanically and chemically, therefore the immiscible approximation described above is not justified a-priori. Nevertheless, to first approximation and on relatively short time scales (hours to days), chemical diffusion among interacting magmas at the plumbing system scale can be neglected (e.g. Ruprecht et al., 2008; Garg et al., 2019), and magmas can be considered immiscible. Here we describe exemplary buoyancy-driven interaction among two natural silicate melts (Figure 6). Density and viscosity of the two melts are computed a-priori using the MagmaFOAM utility `Test-magmaThermoMixture`. As a test case, we reproduce at small scale a typical (Garg et al., 2019) interaction among a volatile-rich basalt ($X_{H_2O} = 2\,wt\%$) and a chemically more evolved andesitic melt. Temperature is set to $T = 1300\,°C$ and pressure is atmospheric. Melt compositions are reported in Table D1. The composition and p, T conditions are considered only in the pre-processing to compute the density and viscosity of the melt that remain constant throughout the simulation. The relevant dimensionless numbers are now $A = 0.0167982$ and $Re = 54.065$ for a physical domain 1 m x 4 m. Surface tension is again neglected. Compared to the previous simulations (e.g., Figure 4), the two liquids have now different kinematic viscosities. The larger viscosity ratio requires to increase significantly the numbers of iterations ($\approx 300$) needed to solve pressure-velocity coupling (keeping $Co_{max} = 0.5$). As a result, the simulation is computationally much more demanding. The simulated time covers the entire overturning process (Figure 6).

### 3.1.2 Rising bubble dynamics

We consider a gas bubble rising in a basaltic melt. The bubble, initially at rest, rises due to buoyancy assuming a variety of shapes depending on the system parameters (e.g., liquid viscosity, surface tension, density contrast). Samkhaniani et al. (2012) demonstrated the ability of `interFoam` to reproduce the different bubble shapes reported in the Grace diagram (Grace, 1973). Our contribution here focuses on the validation of the solver for bubble stability in magmas, through comparison with Suckale et al. (2010a) that used a different numerical method (Figure 7). In this set of 2D simulations, the main goal is to investigate the ability of the solver to reproduce the breakage of a bubble in relation with the shape that it may assume during its rise. Breakage may occur because of the small, random perturbations that form at the melt-bubble interface. No-slip boundary conditions are used for top and bottom boundaries and periodic conditions for the sides. In volcanic context, two parameters

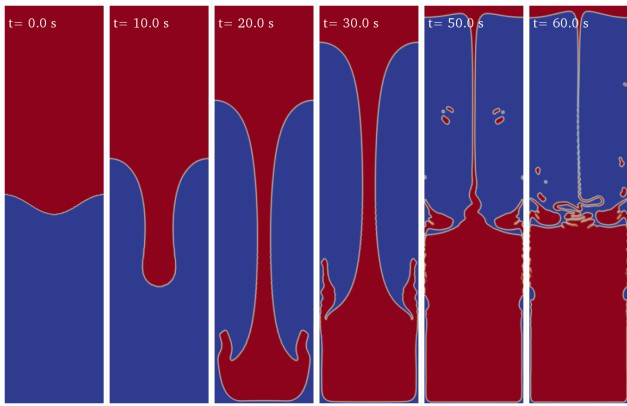

**Figure 6.** Rayleigh-Taylor instability between a volatile-rich ($X_{\mathrm{H_2O}} = 2\,\mathrm{wt\%}$) basalt (bottom) and andesite (top) computed with OpenFOAM solver `interFoam` (color-coded). The physical domain size is $1\,\mathrm{m} \times 4\,\mathrm{m}$.

change significantly with respect to water experiments Samkhaniani et al. (2012): the density $\rho$ and the viscosity $\mu$ of the liquid. While for water $\rho = 10^3\,\mathrm{kg/m^3}$ and $\mu = 10^{-3}\,\mathrm{Pa\,s}$, even a low-viscosity silicate melt (e.g., basalt) has viscosity values of order 10-100 Pa s and the density is above $2500\,\mathrm{kg/m^3}$. Surface tension is $\sigma = 0.073\,\mathrm{N/m}$ for water, while a reasonable value for magmas is $\sigma = 0.15\,\mathrm{N/m}$ (Colucci et al., 2016). In our tests we set $\sigma = 0.3\,\mathrm{N/m}$ and $\rho = 3500\,\mathrm{kg/m^3}$ to be consistent with Suckale et al. (2010a). With $a$ being the bubble radius and $v_0$ the rise velocity, the relevant non-dimensional numbers derived

directly from the governing equations for an incompressible melt are (Suckale et al., 2010a): Reynolds number $\mathrm{Re} = \frac{\rho v_0 a}{\mu}$ (inertial to viscous forces), Froude number $\mathrm{Fr} = \frac{v_0}{\sqrt{ga}}$ (inertia to buoyancy forces), Weber Number $\mathrm{We} = \frac{\rho v_0^2 a}{\sigma}$ (inertia to surface tension) and gas to liquid viscosity ratio $\Pi = \mu_g/\mu$. We can also define the Eötvös number (Eo), which is a combination of Fr and We (Eo = FrWe). Let us note that to be consistent with Suckale et al. (2010a) all non-dimensional numbers here are based on the bubble radius instead of the bubble diameter, which is also commonly used in the literature (e.g. Roghair et al.

(2011)). Considering a constant $\Pi = 10^{-6}$, bubble regimes can be classified using only two adimensional numbers, $\mathrm{Re}$ and $\mathrm{Eo}$. The Reynolds and Eotvos numbers control bubble stability, deformation and breakup. Indicatively, for $\mathrm{Eo} < 1$ and $\mathrm{Re} < 1$ the bubble is stable and preserves its initial spherical shape even in the presence of small perturbations of its interface. For $\mathrm{Eo} > 1$ and $\mathrm{Re} < 1$ the bubble deforms and may breakup if random perturbations affect significantly its surface, while for $\mathrm{Eo} > 1$ and $\mathrm{Re} > 1$ breakup occurs invariably.

Overall, breakup mechanisms are well reproduced in our simulations and bubble shapes at given non-dimensional times are consistent with those reported by Suckale et al. (2010a) for similar values of the non-dimensional numbers (Figure 7) and similar space resolution (20 cells per bubble radius). For the no breakup regime (Figure 7a), the bubble shape in our simulation displays two additional thin wings. In the gradual breakup regime (Figure 7b) small droplets are formed at the rear of the bubble. The results are reported here with higher resolution (40 cells per bubble radius), since with lower resolution the bubble

presents a slightly different shape with a flatter head. In the catastrophic breakup regime (Figure 7c), the bubble immediately collapses forming a large number of small to medium sized bubbles.

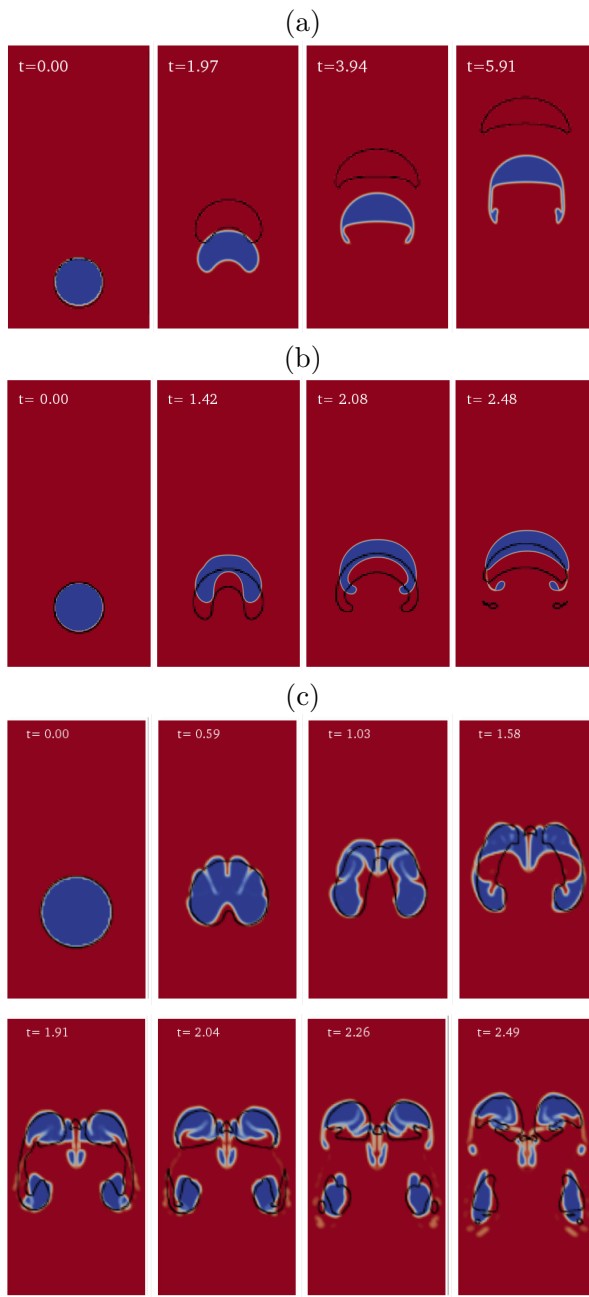

**Figure 7.** Simulation of bubble rise in a basaltic melt using `interFoam` are compared with the results of Suckale et al. (2010a) (black lines) for three different regimes: (a) No breakup (Re $\approx 5$, Fr$^2 \approx 0.4$, We $\approx 90$, and $\Pi = 10^{-6}$), (b) Gradual breakup (Re $\approx 25$, Fr$^2 \approx 0.3$; We $\approx 800$, and $\Pi = 10^{-6}$); (c) Catastrophic breakup (Re $\approx 250$, Fr$^2 \approx 0.16$, We $\approx 1350$, and $\Pi = 10^{-6}$). For each regime, snapshots at different non-dimensional times are shown. To be consistent with Suckale et al. (2010a) here we use the square of the Froude number (Fr$^2 = \frac{v_0^2}{ga}$).

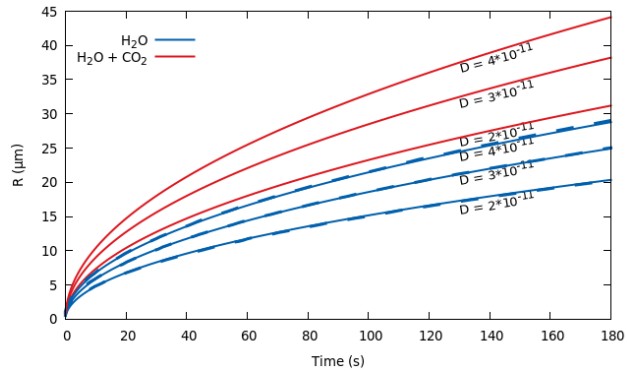

**Figure 8.** Temporal evolution of bubble radius for an instantaneous decompression from $p_0 = 150\,\text{MPa}$ to $p_L = 120\,\text{MPa}$. In blue the comparison between the MagmaFOAM model `multiComponentODERPShellDStatic` (solid lines) and numerical solutions from Lyakhovsky et al. (1996) (dashed lines) that practically coincide for three different values of diffusion coefficient of $H_2O$. The red lines represent the same simulations with $1\text{wt}\%$ of $CO_2$ added in the melt. The diffusion coefficient of $CO_2$ is one order of magnitude smaller than $H_2O$. Initial conditions and parameters (see Appendix B) in all simulations are : $\rho_L = 2300\,\text{kg/m}^3$, $\mu = 5 \cdot 10^4\,\text{Pa s}$, $\sigma = 0.06\,\text{N/m}$, $T = 1123\,\text{K}$, $p_G(t=0) = p_0 + 2\sigma/R(t=0)$, $R(t=0) = 10^{-7}\,\text{m}$, $S_0 = 2 \cdot 10^{-4}\,\text{m}$, $C_{H_2O}^0 = 0.053\text{wt}\%$. Saturation concentration is computed using SOLWCAD (Papale et al., 2006).

### 3.2 Diffusive bubble growth

Here we demonstrate the ability of the ODE solver `multiComponentODERPShellDStatic` to simulate bubble growth in a rhyolitic melt by expansion and mass diffusion. Our solution has been benchmarked by comparison with Lyakhovsky
et al. (1996) for the diffusive growth of water gas bubbles under instantaneous decompression of a hydrated rhyolitic melt. To reproduce the reference solution we assumed a quasi-static diffusion in the liquid shell around the bubble. The quasi-static approximation holds when diffusion is fast relative to decompression rate (Lensky et al., 2004). The reference solutions for three different values of the diffusion coefficient are well reproduced by our model (Figure 8). We repeat the same simulations with the addition of $1\%$ of $CO_2$ (red lines in Figure 8). The multicomponent saturation surface is calculated using SOLWCAD
(Papale et al., 2006). The bubble radius increases by $\approx 50\%$ and the gas volume fraction triplicates (see Equation (B6)).

### 3.3 Eulerian multi-fluid modeling

In this section we test the ability of the OpenFOAM two-fluid eulerian solver `reactingTwoPhaseEulerFoam` to deal with flow problems with a large number of small (unresolved) gas bubbles dispersed into a liquid phase. `reactingTwoPhaseEulerFoam`, coupled with the magmaFOAM libraries, is tested in problems involving multiphase shock tubes as well as by simulating a
multiphase-multicomponent reacting box.

### 3.3.1 Shock Tube Simulations

Decompression of a pressurized bubbly magma is a common trigger of explosive volcanic eruptions (e.g., Gonnermann and Manga, 2007). When a high-pressure magma reservoir is decompressed, a shock wave and a contact wave propagate into the low pressure region, typically the atmosphere, and a rarefaction wave propagates into the bubbly magma (Koyaguchi and Mitani, 2005; La Spina et al., 2017), akin to shock tube devices. The latter have been extensively used to study wave propagation phenomena in compressible fluids. Usually high and low pressure regions are separated by a diaphragm, the instantaneous removal of which initiates highly transient dynamics (Stadtke, 2006). Assuming strictly one-dimensional flow conditions (i.e., ignoring the effects of shear viscosity), the shock tube mathematically represents a Riemann problem where the initial velocities on both sides have been set to zero. For the specific case of ideal equation of state, an analytical solution can be derived for a pure single phase (Stadtke, 2006). Multiphase flow processes are generally governed by deviations from mechanical and thermal equilibrium between the phases. Nevertheless, the assumptions of homogeneous flow (equal phase velocities) and thermal equilibrium between the phases allow us to define a special limiting case for which a semi-analytical solution can be derived (Stadtke, 2006). We test the `reactingTwoPhaseEulerFoam` solver in inviscid one-dimensional and viscid axisymmetric simulations of single-phase and two-phase shock tubes. Axisymmetric simulations allow us to investigate the effect of melt viscosity on the radial velocity profile, through the Giordano et al. (2008) model. The Lange and Carmichael (1987) equation of state is tested here for the propagation of rarefaction and shock waves.

**Single phase** We perform a standard Sod shock tube benchmark for pure air gas using a perfect gas equation of state. A nearly perfect agreement between the simulation and the analytical solution has been found by discretizing the one-dimensional computational domain with 5000 cells. Then, we test the solver by simulating a shock-tube with pure liquid water using the SPWAT equation of state (Stadtke, 2006) implemented in MagmaFOAM. Discretizing the computational domain with the same number of cells, the contact and shock wave discontinuities are well resolved and do not display any instabilities. Finally, we perform a shock-tube simulation with pure liquid basalt (Table D1) using the equation of state for silicate melts proposed by Lange and Carmichael (1987) and implemented in MagmaFOAM. We use the same computational domain and the same numerical schemes used in the liquid water test. Across the shock discontinuity the solution is more diffusive compared to the test for liquid water, while the contact discontinuity is still well resolved. The Figures for the single phase shock tubes described above are reported in Appendix C.

**Two-phase** We perform two-phase shock tube simulations for gas air-liquid water and gas water-liquid basalt (Table D1) shock tubes (Figures 9, 10, 11). The equations of state for each phase are the same as for the single-phase cases. In all the simulations, the size of the dispersed phase (i.e., gas bubbles or liquid droplets), instead of being determined by a proper model (i.e., bubble growth model), is kept constant and used as a tuning parameter. This unphysical assumption allows us to control the mechanical and thermal disequilibrium between the gas and liquid phases in order to compare the simulation with the limiting analytical solution for a homogeneous flow (Stadtke, 2006). It is worth noting that, even if the size of the dispersed phase is kept constant, its volume fraction can change according to the mass conservation equations.

First, we benchmark the solver with a gas air-liquid water shock tube for which a limiting analytical solution is provided (Stadtke, 2006) (Figure 9). Initial gas volume fraction is 0.1 in the high-pressure region (to the left of the interface) and 0.05 in the low-pressure region (to the right of the interface). Overall, we find a remarkable good agreement with the exact solution. The contact and shock wave discontinuities are well resolved and do not display any instabilities. The numerical solution is only slightly diffusive at the onset of the rarefaction wave. The overshoot visible in the velocity is produced by the mechanical decoupling between the liquid and the dispersed gas phase (Stadtke, 2006) and disappears reducing the bubble size, as will be discussed in the next subsection.

The same simulation setup is used to simulate basalt (liquid) - water (gas) shock tube (Figure 10). In this case the simulation is axisymmetric, in order to understand the role of melt viscosity. The shape of axial profiles of pressure, velocity, gas volume fraction and mixture density are similar to 1D shock tube (Figure 9) for the air-water system. Velocity profiles along radial coordinate are flat with a narrow boundary layer near the walls. In this case the higher viscosity ($\approx 10\,\mathrm{Pa\,s}$) drastically reduces the mechanical phase decoupling and the phase velocities are superimposed.

Finally, we use the same simulation setup of Figures 9 and 10, but with an initial gas volume fraction in the low pressure region (to the right of the interface) equal to 1 (Figures 11 and C4 in Appendix C). This configuration is more appropriate for a volcanic application where the shock wave travels in the atmosphere. In this case the continuous and dispersed phases can invert, thus the bubbly flow, where bubble are dispersed in the continuous liquid phase, becomes a particle flow, where the liquid droplets are dispersed in the gas. This process, usually called fragmentation in volcanological literature, can be modelled, as a first approximation, using a critical volume fraction criterion ($0.5 < \alpha < 0.7$; e.g. Sparks 1978 or La Spina et al. 2017). When the rarefaction wave propagates into the high pressure region (i.e., left side), the bubbly magma expands, accelerates and fragments. Due to the higher compressibility of the gas phase compared to the liquid melt, the temperature subplot shows phase decoupling during expansion, the amount of which depends on the adopted heat transfer model.

**The phase coupling problem** Even if we limit to bubbly flow regimes, magmatic system are characterized by a wide range of viscosities (from $0.1\,\mathrm{Pa\,s}$ to $10^9\,\mathrm{Pa\,s}$) and bubble sizes (from few microns to decimeters). The bubble relaxation time ($\tau_b$) is directly proportional to the square of the bubble diameter and inversely proportional to the kinematic viscosity of the continuous liquid phase ($\tau_b \propto D_b^2/\nu_l$). In magmatic phenomena, when considering small bubbles (e.g., $100\,\mu\mathrm{m}$) and even relatively low viscosities (e.g., $10\,\mathrm{Pa\,s}$), $\tau_b$ can reach very small values ($\tau_b \approx 10^{-6}\,\mathrm{s}$), resulting in very strong mechanical phase coupling. Numerical algorithms like the one implemented in OpenFOAM, based on segregated solvers, require special care in order to ensure convergence of the solution when phase coupling is tight (Karema and Lo, 1999). The Partial Elimination Algorithm (PEA), implemented in OpenFOAM, shows the best convergence performance compared to other algorithms (Karema and Lo, 1999; Venier et al., 2016). Here we test the PEA method for shock tube simulations conditions within the range of interest for magmatic applications. In Figure 12 we compare the analytical solution to the simulation results for different values of $\tau_b$ obtained by changing the bubble diameter. Decreasing $\tau_b$ from $10^{-1}\,\mathrm{s}$ to $10^{-3}\,\mathrm{s}$ the velocities of the two phases tend to overlap as expected, and agree with the homogeneous analytical solution. However, by further decreasing $\tau_b$ the solution diverges even when increasing 40 times the number of corrector cycles. This is an important limitation in the use of the multi-fluid solver. A possible workaround, currently under investigation, is to implement a limiter for the relaxation time.

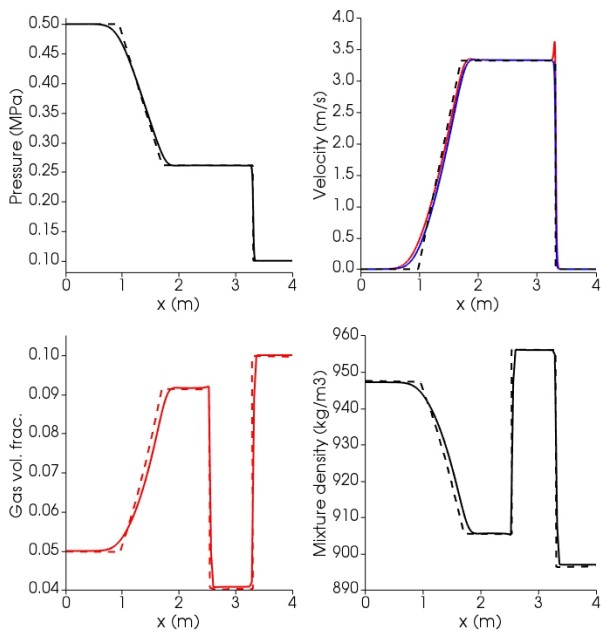

**Figure 9.** Results at time $t = 0.015\,\text{s}$ for the air-water shock tube using the SPWAT EOS. Solid lines: OpenFOAM; dashed lines: analytical solution (Stadtke, 2006); Black lines: mixture; Blue lines: liquid (water); Red lines: gas (air). Mixture density is calculated a posteriori as $\rho_{mix} = \alpha_g \rho_g + (1 - \alpha_g)\rho_l$, where l is liquid and g is gas. At time 0, the interface dividing high pressure (left, l) from low pressure (right, r) zone is placed at 2.5 m. Initial conditions: $P_l = 0.5\,\text{MPa}$, $P_r = 0.1\,\text{MPa}$; $T_l = T_r = 300\,\text{K}$ for both phases; gas volume fraction $\alpha_l = 0.05$, $\alpha_r = 0.1$; $U_l = U_r = 0$ for both phases. Isobaric heat capacities of gas air and liquid water are, respectively, $Cp_g = 1004.5\,\text{Jkg}^{-1}\text{K}^{-1}$ and $Cp_l = 4195\,\text{Jkg}^{-1}\text{K}^{-1}$ (https://webbook.nist.gov/). Prandtl numbers of air and water are, respectively, 0.7 and 2.289, corresponding to thermal conductivities of $0.02\,\text{W K}^{-1}\text{m}^{-1}$ and $0.67\,\text{W K}^{-1}\text{m}^{-1}$ (https://webbook.nist.gov/).

### 3.3.2 Reacting box

A many-bubble system at zero gravity where bubbles grow by mass diffusion is analyzed here. The liquid phase is a basaltic melt (Table D1) with dissolved water and carbon dioxide whose properties are modelled by the Lange and Carmichael (1987) equation of state and the rheological equation of Giordano et al. (2008). The ideal gas equation of state has been used for the gas phase. As this is a many-bubble system, bubble growth is approximated through a subgrid model (see section 2.2). The mass transfer coefficient (i.e., $k_i$ in Equation (5)) is calculated according to a spherical model and the heat transfer coefficient

according to spherical and Ranz-Marshall models, both already implemented in OpenFOAM. In addition, the I group IATE model (Ishi and Hibiki, 2006) is used to compute the interfacial area required by mass and heat transfer rate. The Interfacial Area Transport Equation (i.e., IATE) is a fundamental equation, formulated from the Boltzmann transport equation, describing the change of surface area between phases, assuming spherical shape of the dispersed phase (Ishi and Hibiki, 2006).

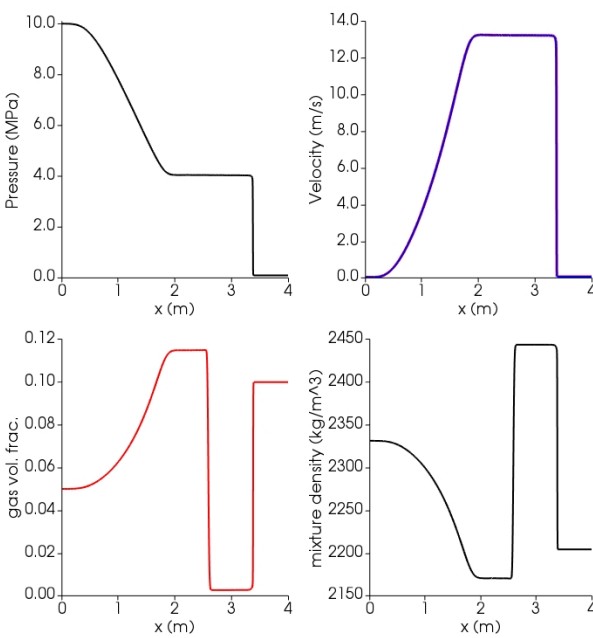

**Figure 10.** Results at time $t = 0.0065\,\mathrm{s}$ for the axisymmetric gas water-basalt shock tube using the Lange-Carmichael EOS (Lange and Carmichael, 1987) and the viscosity model of Giordano et al. (2008). Blue lines: liquid (basalt); Red lines: gas (water). At time 0, the interface dividing high pressure (left, l) from low pressure (right, r) zone is placed at 2.5 m. Initial conditions: $P_l = 10\,\mathrm{MPa}$, $P_r = 0.1\,\mathrm{MPa}$; $T_l = T_r = 1273\,\mathrm{K}$ for both phases; gas volume fraction $\alpha_l = 0.05$, $\alpha_r = 0.1$; $U_l = U_r = 0$ for both phases. Isobaric specific heat capacities in the gas and liquid phase are, respectively, $C_{P_g} = 2510\,\mathrm{J/kg^{-1}K^{-1}}$ (https://webbook.nist.gov/) and $C_{P_l, H_2O} = 2278\,\mathrm{J/kg^{-1}K^{-1}}$, $C_{P_l, basalt} = 1600\,\mathrm{J/kg^{-1}K^{-1}}$ (Lesher and Spera, 2015). Thermal conductivity of the liquid is $1.5\,\mathrm{W\,K^{-1}m^{-1}}$ (Lesher and Spera, 2015); for the water gas phase a Prandtl number of 0.9 is used, corresponding to a thermal conductivity of $0.14\,\mathrm{W\,K^{-1}m^{-1}}$ (https://webbook.nist.gov/)

.

At time zero, a small amount of gas is uniformly distributed in the box and the liquid-gas system is out of thermodynamic equilibrium because the liquid is oversaturated in both $H_2O$ and $CO_2$. After $\approx 4 \cdot 10^5$ s, $H_2O$ has reached the thermodynamic equilibrium increasing the gas volume fraction to $\approx 33\%$ (Figure 13) and the bubble size increased from 1 cm to about 4 cm (not shown in the Figure). This time is consistent with the time scale that characterizes diffusive mass transfer of $H_2O$ (diffusion coefficient $D = 10^{-9}\,\mathrm{m^2/s}$, Baker et al., 2005) around a bubble with radius $R \approx 2\,\mathrm{cm}$ ($\tau_d = R^2/D$; Lensky et al., 2004). The dissolved $CO_2$ is still out of thermodynamic equilibrium, as expected, because the diffusion coefficient is lower, being set to $D = 10^{-10}\,\mathrm{m^2/s}$ (Baker et al., 2005). The density and viscosity of the liquid vary with the decreasing dissolved $H_2O$. The gas density decreases because of increasing $H_2O$ and decreasing $CO_2$ that produce a decrease of the molar mass of the gas mixture.

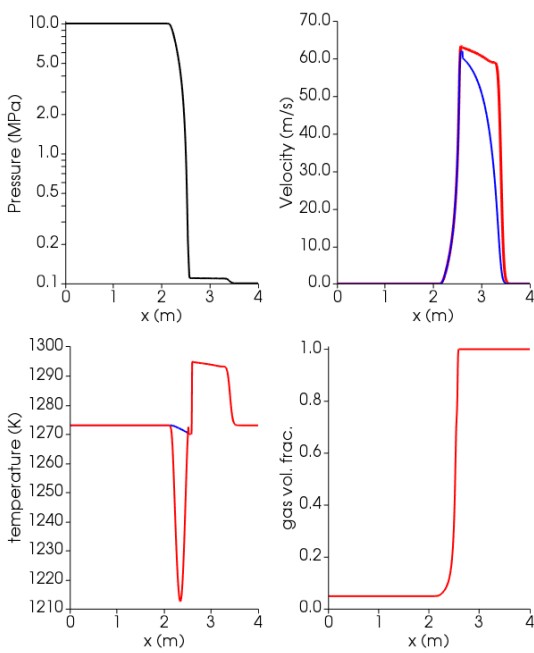

**Figure 11.** Results at time $t = 0.001\,\mathrm{s}$ for the axsymmetric gas water-basalt shock tube using the Lange-Carmichael EOS (Lange and Carmichael, 1987) and the viscosity model of Giordano et al. (2008), with liquid phase switching from continuous to dispersed. Blue lines: liquid (basalt); Red lines: gas (water). At time 0, the interface dividing high pressure (left, l) from low pressure (right, r) zone is placed at 2.5 m. Initial conditions: $P_l = 10\,\mathrm{MPa}$, $P_r = 0.1\,\mathrm{MPa}$; $T_l = T_r = 1273\,\mathrm{K}$ for both phases; gas volume fraction $\alpha_l = 0.05$, $\alpha_r = 1$; $U_l = U_r = 0$ for both phases. Isobaric specific heat capacities in the gas and liquid phase are, respectively, $C_{\mathrm{P_g}} = 2510\,\mathrm{J/kg^{-1}K^{-1}}$ (https://webbook.nist.gov/) and $C_{\mathrm{P_{l,H_2O}}} = 2278\,\mathrm{J/kg^{-1}K^{-1}}$, $C_{\mathrm{P_{l,basalt}}} = 1600\,\mathrm{J/kg^{-1}K^{-1}}$ (Lesher and Spera, 2015). Thermal conductivity of the liquid is $1.5\,\mathrm{W\,K^{-1}m^{-1}}$ (Lesher and Spera, 2015); for the water gas phase a Prandtl number of 0.9 is used, corresponding to thermal conductivity of $0.14\,\mathrm{W\,K^{-1}m^{-1}}$ (https://webbook.nist.gov/).

## 4 Conclusions

In this work we have presented MagmaFOAM, a library based on OpenFOAM that contains dedicated tools for the simulation of multiphase flows in magmatic systems. The MagmaFOAM implementation results in a flexible framework which is ideal for development, testing, coupling and application of the great collection of existing and future modeling strategies needed to tackle the variety of non linear multi-scale processes characterizing magma flows. MagmaFOAM includes dedicated multi-component constitutive models for dealing with realistic properties for silicate melt-gas systems as well as different utilities that automatize pre- and post-processing operations. We have analyzed a number of test cases that illustrate the current capabilities and limitations of different modeling approaches in solving well-defined and reproducible flow problems, setting a solid ground for future model selection, improvement and inter-comparison studies. We have shown some of the ingredients that can be used for simulating the interaction among different silicate melts, as well as between melts and fluid phases, under different assumptions and aimed at different portions of the magmatic system (deep reservoirs vs. shallow conduits). Applica-

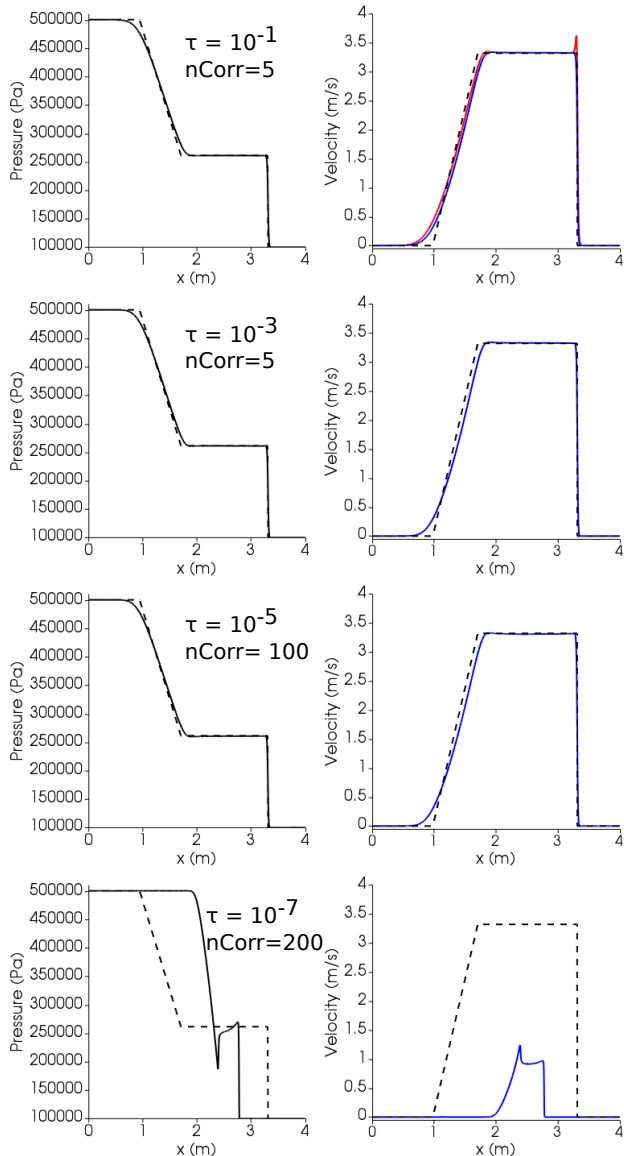

**Figure 12.** Air-water shock tube simulations at different relaxation times $\tau$ and number of correctors. Dashed lines: analytical solution; Solid lines: simulation. Blue lines: liquid (water); Red lines: gas (air). At time 0, the interface dividing high pressure (left, l) from low pressure (right, r) zone is placed at 2.5 m. Initial conditions: $P_l = 0.5\,\mathrm{MPa}$, $P_r = 0.1\,\mathrm{MPa}$; $T_l = T_r = 300\,\mathrm{K}$ for both phases; gas volume fraction $\alpha_l = 0.05$, $\alpha_r = 0.1$; $U_l = U_r = 0$ for both phases. Isobaric heat capacities of gas air and liquid water are, respectively, $C_{P_g} = 1004.5\,\mathrm{J/kg^{-1}K^{-1}}$ and $C_{P_l} = 4195\,\mathrm{J/kg^{-1}K^{-1}}$ (https://webbook.nist.gov/). Prandtl numbers of air and water are, respectively, 0.7 and 2.289, corresponding to thermal conductivities of $0.02\,\mathrm{W\,K^{-1}m^{-1}}$ and $0.67\,\mathrm{W\,K^{-1}m^{-1}}$, resepctively (https://webbook.nist.gov/).

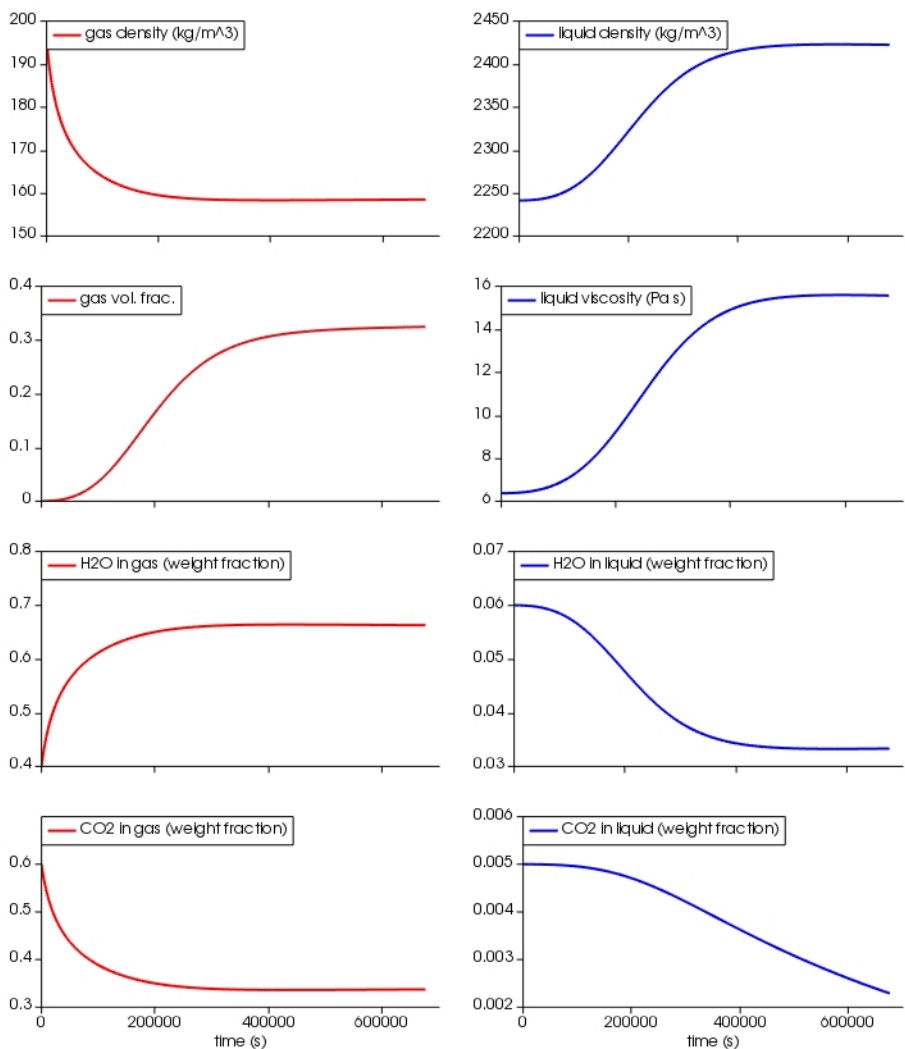

**Figure 13.** Reacting box simulation. At time 0 a small amount of gas (volume fraction $\alpha_g = 10^{-4}$) is uniformly distributed in the box, the basaltic melt is oversaturated in $H_2O$ (5 wt%) and $CO_2$ (5000 ppm) at 80 MPa and 1373 K. The diffusion coefficients for $H_2O$ and $CO_2$ in the basalt are, respectively, $10^{-9}\,m^2/s$ and $10^{-10}\,m^2/s$ (Baker et al., 2005). Isobaric specific heat capacities in the gas and liquid phase are, respectively, $C_{P_{g,H_2O}} = 2900\,J/kg^{-1}K^{-1}$, $C_{P_{g,CO_2}} = 1390\,J/kg^{-1}K^{-1}$ (https://webbook.nist.gov/, Beaton et al., 1987) and $C_{P_{l,H_2O}} = 2278\,J/kg^{-1}K^{-1}$, $C_{P_{l,CO_2}} = 1600\,J/kg^{-1}K^{-1}$, $C_{P_{basalt}} = 1600\,J/kg^{-1}K^{-1}$ (Lesher and Spera, 2015). Thermal conductivity of the liquid is $1.5\,W\,K^{-1}m^{-1}$ (Lesher and Spera, 2015); for the gas phase Prandtl numbers of 0.9 for $H_2O$ and 0.7 for $CO_2$ are used, corresponding to thermal conductivities of $0.16\,W\,K^{-1}m^{-1}$ and $0.09\,W\,K^{-1}m^{-1}$ (https://webbook.nist.gov/, Beaton et al., 1987).

tions of MagmaFOAM can thus include the study of magma mingling and mixing, as well as slug rising dynamics, or volatile
flushing. Nevertheless, important limitations remain, most notably the development of a magma-specific mixture approach; or
the intrinsic complications in modeling the transition from tight to loose phase coupling (Section 3.3.1).

The framework described in this work allows for maximum flexibility and adaptability, giving the possiblity to explore
magmatic systems with different approaches given the specific conditions aimed at. As an example, the MagmaFOAM modular
approach allows the coupling of its bubble growth models with both single and multi-fluid solvers, Lagrangian particle tracking,
or with more complex constitutive equations. Indeed, at different stages within the evolution of magmatic plumbing systems,
different modeling approaches can be more appropriate to capture the fundamental physics governing the dynamics: while
low-gas-fraction, deep reservoirs may well be approximated by mixture theory, at shallower levels phase decoupling becomes
important and multi-fluid descriptions are more appropriate.

The tool is meant to be under continuous development, already underway. The addition of population balance equations to
single and multi fluid models to statistically describe the dispersed phases (bubbles and crystals, Marchisio and Fox (2013))
will improve our understanding of how polydispersity can impact magmatic system evolution (Colucci et al., 2017a; de'
Michieli Vitturi and Pardini, 2020). In large-scale multi-fluid simulations, the exchanges of mass, momentum, and energy
through the interface between phases need to be modelled accurately to determine the rate of phase change and the degree
of mechanical and thermal disequilibrium between phases. The population balance is a statistical approach for modelling
the mesoscale dynamics, widely used in chemical engineering, which describes the temporal and space evolution of a large
number of particles through a number distribution function (Yeoh and Tu, 2019a). In this way microscopic processes involving
bubble dynamics and interactions between bubbles can be included in large-scale multifluid simulations. In fact, DNS allows
to model particle-particle interactions and capture emerging behaviours in complex systems; however, the large quantity of
microphysics taken into account in DNS has to be filtered and condensed in a sub-model to be used in large-scale simulations.
Mesoscopic models represent intermediate models that describe, through a set of mesoscale variables, the microphysics of the
system. The formulation of population balance requires adequate closure models for the microphysics that can be developed
with the aid of experimental (Mancini et al., 2016) and DNS investigations (Marchisio and Fox, 2013). The inclusion of
Lagrangian tracers will result in a more detailed description, with respect to multi fluid models, of the micro-physics that
determines the macroscopic properties driving the dynamics. In the Eulerian-Lagrangian approach, bubbles are treated as
discrete Lagrangian particles in an ambient Eulerian continuous flow. (e.g. Ghahramani et al., 2019). This approach in fact is
more appropriate than multi fluid models when the number of particles is too small to be treated as a continuum, or when single
particles' behaviour (e.g. rapidly expanding/contracting bubbles) is so specific that they are not well represented by unique
averaged fields density, velocity or temperature (e.g. Ghahramani et al., 2019). With respect to the DNS approach, where
bubble-bubble and bubble-melt interactions emerge self-consistently, in the Eulerian-Lagrangian models phase interactions are
defined by constitutive models. However, the Eulerian-Lagrangian approach, compared to the DNS, allows to simulate larger
populations of particles at a much lower computational cost. The study of complex mixing behaviour in magma mushes is
only an example of possible applications (Bergantz et al., 2015). Finally, engineering applications have benefited from models
that combine different approaches, e.g. interface resolving and subgrid dispersed phase modeling with single or multi fluid

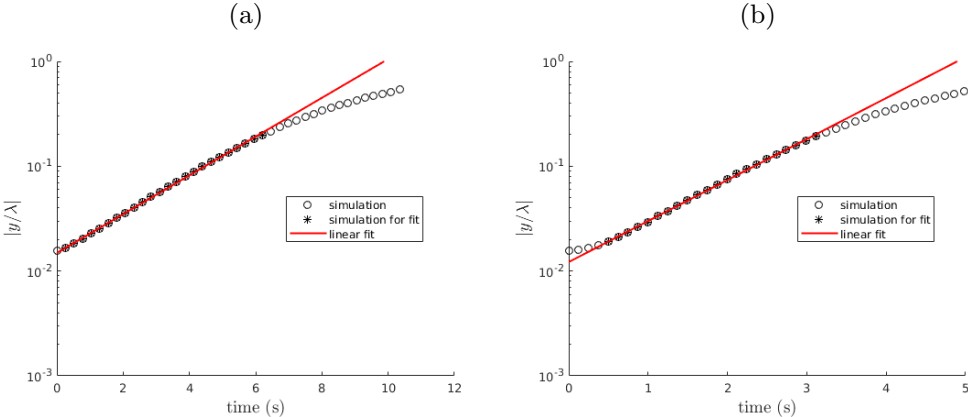

**Figure A1.** Time evolution of the amplitude of two single mode perturbations ($k = 0.5$ (a), and $k = 0.15$ (b)) for the linear Rayleigh Taylor instability benchmark. The growth rate of the perturbation is extrapolated with a linear regression excluding data in late (physical) and eventually early (spurious) phases characterised by non linear effects (data not marked with an asterisk).

frameworks. These hybrid models, although not fully mature yet, allow in principle modeling at the same time the broad range
of interface scales that typically characterize gas-liquid flows including regime transitions (e.g., Wardle and Weller, 2013). From a volcanological perspective, predicting flow regime changes is of crucial importance to understand effusive-explosive transitions in eruptive activity (Gonnermann and Manga, 2007).

## Appendix A: Linear Rayleigh Taylor instability

Figure A1b shows how a small wave number perturbation ($k = 0.15$) initially grows with slower non constant growth rate.
Overall this effect make the extrapolated growth rate smaller than expected. However, after a relatively small time interval, the growth rate becomes constant with a value that results to be in good agreement with the theoretical one (Figure A2). This spuriuos effect gradually decreases till it diseappears as the wave number of the perturbation increases (Figure A1a). The simulations are done using the solver `interFoam` with adaptable time step ($Co_{max} = 0.01$).

## Appendix B: multiComponentODERPShellDStatic: model equations

A modified form of the Rayleigh-Plesset equation describes the hydrodynamics of the growth of a multi component spherical bubble in a finite incompressible shell of liquid.

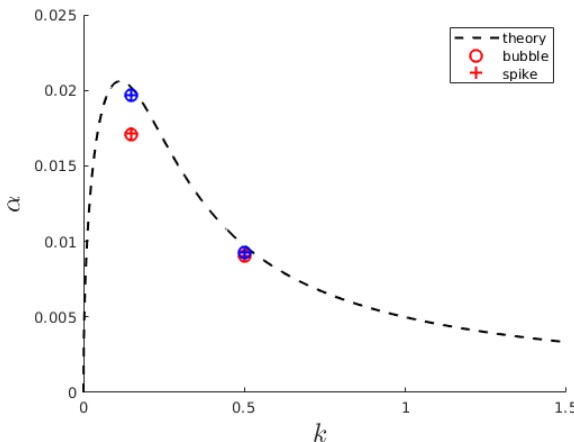

**Figure A2.** Extrapolated growth rate for two perturbations with linear regression excluding (blue) or not (red) data in the initial phase characterised by non linear spurious effects.

$$\rho_L R \frac{d^2 R}{dt^2}\left(1-\frac{R}{S}\right) + \rho_L\left(\frac{dR}{dt}\right)^2\left(\frac{3}{2}-\frac{2R}{S}+\frac{R^4}{2S^4}\right) = \tag{B1}$$

$$p_G - p_L + 4\frac{dR}{dt}R^2\left(-3\int_R^S \frac{\mu(r)}{r^4}dr\right) - \frac{2\sigma}{R}.$$

In the above, $\rho_L$ is the liquid density, $R$ is the bubble radius, $S$ is the radius of the shell ($S^3 = S_0^3 + R^3$; $S_0^3 = S^3(t =$

$0) - R^3(t=0)$), $p_G$ is the gas pressure inside the bubble, $p_L$ is the pressure acting on the outside of the liquid shell, $\sigma$ is the surface tension, $t$ is the time and $\mu$ is the liquid dynamic viscosity that depends on the concentration of dissolved volatiles in the shell. Given $p_L(t)$ this represents an equation that can be solved to find $R(t)$ provided $p_G(t)$. $p_G$ is given by combining the mass conservation of the gas phase with an equation of state for a perfect gas. Mass conservation of the gas phase is given by

$$\frac{d}{dt}\left(R^3 \rho_G\right) = 3R^2 \rho_L \sum_{i=1}^{N} D_i \left[\frac{\partial C_i}{\partial r}\right]_{r=R}, \tag{B2}$$

where $D_i$ is the mass diffusivity and $C_i$ the concentration of the i-th species dissolved in the melt. Mass conservation of the i-th dissolved specie is given by

$$\frac{d}{dt}\left(R^3 \rho_G Y_i\right) = 3R^2 \rho_L D_i \left[\frac{\partial C_i}{\partial r}\right]_{r=R}, \tag{B3}$$

where $Y_i$ is the concentration in the gas phase. Assuming local thermodynamic equilibrium at the bubble-melt interface (i.e., $C_i(R) = C_i^{\text{sat}}(p_G)$, where $C_i^{\text{sat}}$ is the saturation concentration), a zero gradient boundary conditions at the shell boundary and

a quasi-static diffusion in the shell (Lyakhovsky et al., 1996), the term in square brackets in equations (B2) and (B3) is given

by

$$\left[\frac{dC_i}{dr}\right]_{r=R} = \frac{S_0^3\left(C_i^0 - C_i^{\text{sat}}\right) + \frac{1}{\rho_L}\left(\left[R^3\rho_G Y_i\right]_{t=0} - R^3\rho_G Y_i\right)}{S_0^3 R - \frac{3}{2}\left(S^2 - R^2\right)R^2},\tag{B4}$$

where $C_i^0$ is the concentration in the melt at time 0. Assuming constant viscosity, the term in Equation (B1) is analytically integrated to obtain

$$\left(-3\int_R^S \frac{\mu(C_i(r))}{r^4}dr\right) = \mu\left(\frac{1}{S^3} - \frac{1}{R^3}\right).\tag{B5}$$

For a monodispersed distribution, the gas volume fraction is given by

$$\alpha = \frac{R^3}{S^3}.\tag{B6}$$

## Appendix C: Shock Tube

Figures C1, C2 and C3 show results from the single phase shock tube simulations discussed in Section 3.3.1. Figure C4 shows results from the air-water shock-tube with liquid phase switching from conituous to dispersed.

## Appendix D: Magmatic Compositions

Table D1 reports the compositions in terms of major oxides of the magmas used in the simulations shown in the manuscript.

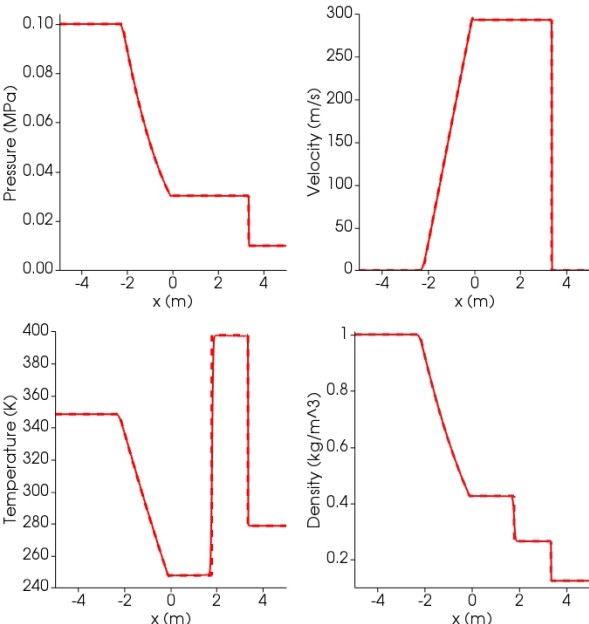

**Figure C1.** Results at time $t = 0.006\,\text{s}$ for the air Sod shock tube. Dashed lines: analytical solution; Solid lines: simulation. At time 0, the interface dividing high pressure (left, l) from low pressure (right, r) zone is placed at 0 m. Initial conditions: $P_l = 0.1\,\text{MPa}$, $P_r = 0.01\,\text{MPa}$; $T_l = 348.432\,\text{K}$, $T_r = 278.746\,\text{K}$; gas volume fraction $\alpha_l = 1$, $\alpha_r = 0$; $U_l = U_r = 0$. Isobaric heat capacity is $C_P = 1004.5\,\text{J kg}^{-1}\text{K}^{-1}$, corresponding to heat capacity ratio $\gamma = 1.4$.

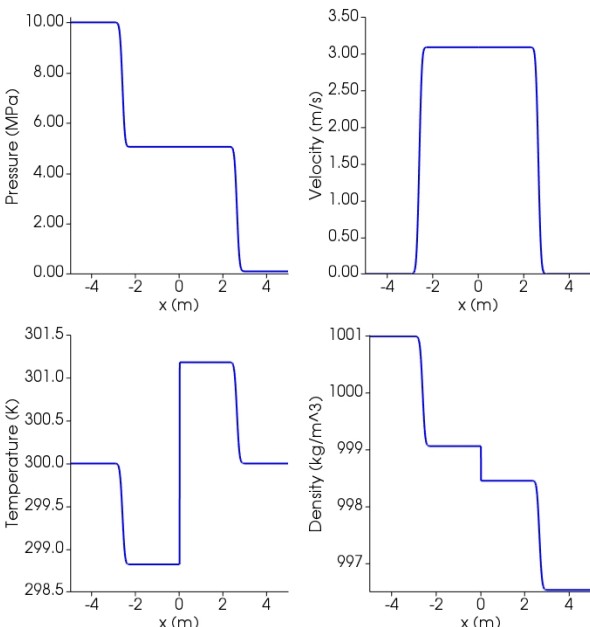

**Figure C2.** Results at time $t = 0.00164$ s for single-phase shock tube with liquid water using SPWAT EOS. At time 0, the interface dividing high pressure (left, l) from low pressure (right, r) zone is placed at 0. Initial conditions: $P_l = 10$ MPa, $P_r = 0.1$ MPa; $T_l = T_r = 300$ K; gas volume fraction $\alpha_l = \alpha_r = 0$; $U_l = U_r = 0$. Isobaric heat capacity is $C_P = 4195$ J kg$^{-1}$K$^{-1}$ (https://webbook.nist.gov/).

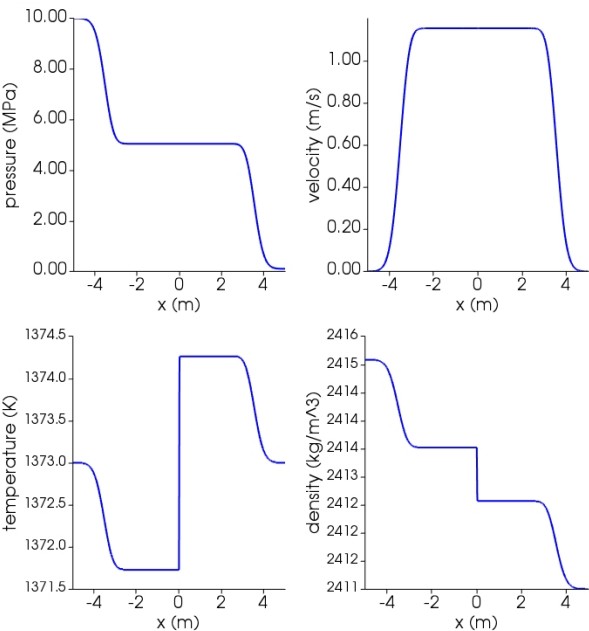

**Figure C3.** Results at time $t = 0.002\,\mathrm{s}$ for single-phase shock tube with basaltic melt using Lange-Carmichael EOS. At time 0, the interface dividing high pressure (left, l) from low pressure (right, r) zone is placed at 0. Initial conditions: $P_\mathrm{l} = 10\,\mathrm{MPa}$, $P_\mathrm{r} = 0.1\,[MPa$; $T_\mathrm{l} = T_\mathrm{r} = 1373\,\mathrm{K}$; gas volume fraction $\alpha_\mathrm{l} = \alpha_\mathrm{r} = 0$; $U_\mathrm{l} = U_\mathrm{r} = 0$. Isobaric heat capacity is $C_\mathrm{P} = 1600\,\mathrm{J\ kg^{-1}K^{-1}}$ (Lesher and Spera, 2015). Thermal conductivity is $1.5\,\mathrm{W\ K^{-1}m^{-1}}$ (Lesher and Spera, 2015).

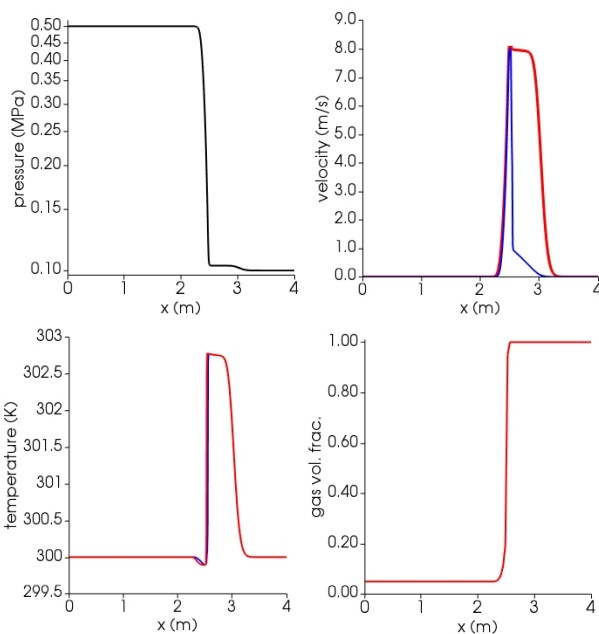

**Figure C4.** Results at time $t = 0.0015\,\mathrm{s}$ for the air-water shock tube with liquid phase switching from continuous to dispersed. Blue lines: liquid (water); Red lines: gas (air). At time 0, the interface dividing high pressure (left, l) from low pressure (right, r) zone is placed at 2.5 m. Initial conditions: $P_\mathrm{l} = 0.5\,\mathrm{MPa}$, $P_\mathrm{r} = 0.1\,\mathrm{MPa}$; $T_\mathrm{l} = T_\mathrm{r} = 300\,\mathrm{K}$ for both phases; gas volume fraction $\alpha_\mathrm{l} = 0.05$, $\alpha_\mathrm{r} = 1$; $U_\mathrm{l} = U_\mathrm{r} = 0$ for both phases. Isobaric heat capacities of gas air and liquid water are, respectively, $C_{\mathrm{P_g}} = 1004.5\,\mathrm{J\,kg^{-1}K^{-1}}$ and $C_{\mathrm{P_l}} = 4195\,\mathrm{J\,kg^{-1}K^{-1}}$ (https://webbook.nist.gov/). Prandtl numbers of air and water are, respectively, 0.7 and 2.289, corresponding to thermal conductivities of $0.02\,\mathrm{W\,K^{-1}m^{-1}}$ and $0.67\,\mathrm{W\,K^{-1}m^{-1}}$ (https://webbook.nist.gov/).

**Table D1.** Oxides composition for the magmas used in benchmarking simulations. Amounts are relative.

|          | $SiO_2$ | $TiO_2$ | $AlO_2$ | $Fe_2O_3$ | FeO    | MnO    | MgO    | CaO    | $Na_2O$ | $K_2O$ |
|----------|---------|---------|---------|-----------|--------|--------|--------|--------|---------|--------|
| andesite | 0.587   | 0.0088  | 0.1724  | 0.0331    | 0.0409 | 0.0014 | 0.0337 | 0.0688 | 0.0353  | 0.0164 |
| basalt   | 0.484   | 0.0167  | 0.178   | 0.0186    | 0.0836 | 0.0018 | 0.0553 | 0.102  | 0.0387  | 0.0211 |

*Code and data availability.* The version of the model used to produce the results shown in this paper, as well as input data and scripts to replicate all the simulations presented in this paper, are archived on Zenodo (Brogi et al., 2021).

*Author contributions.* FB and SC developed and tested the software, including pre- and post-processing, performed the simulations and wrote the first draft of the paper. CPM contributed to paper writing and supervised the work. JM worked on bubble dynamics and its analysis. MdMV provided guidance and help on the multifluid solver. PP provided supervision and reviewed the original draft.

*Competing interests.* The Authors declare no competing interests.

*Acknowledgements.* This work has been supported by Istituto Nazionale di Geofisica e Vulcanologia (INGV) and Istituto Nazionale di
Oceanografia e Geofisica Sperimentale (OGS) under HPC-TRES program award no 2016-05 to F. Brogi. This research has received funding from European Union's Horizon 2020 research and innovation programme under the EUROVOLC project, grant agreement no 731070, and under the ChEESE project, grant agreement no 823844; and from Italy's MIUR PRIN grant 2015L33WAK. The authors thank Jenny Suckale and an anonymous reviewer for their constructive comments that considerably improved the manuscript.

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
