# Peer review of "MagmaFOAM-1.0: a modular framework for the simulation of magmatic systems"

_Geoscientific Model Development, 2021_

## Referee Comment (RC1)

**Comments to: "MagmaFOAM-1.0: a modular framework for the simulation of magmatic systems" by Brogi et al.**

**Manuscript submitted for publication on GMD**

**Description**

This work describes a software library, which extends OpenFOAM, dedicated to the solution of problems typically encountered in modeling volcanic processes.

**General comments**

The library contains already existing subroutines (eg: SOLWCAD) and new interesting software. However, the existing software has been recoded within the same modular framework (OpenFOAM), allowing an easier use by model developer.
I have appreciated the efforts of the authors and I think that MagmaFOAM will be very useful for the volcanological community.

**Minor comments**

- Of course, eq.(3) is valid for $T > C$

- Line 256: The Reynolds number is more often based on the bubble diameter that bubble radius.

- Line 257: The definition of the Weber number seems incorrect. It should contain the surface tension in the denominator. Please check.

- The definition of $\Pi = \mu/\mu_g$ at line 258 seems incompatible with its value ($10^{-6}$) reported at line 259 and in the caption of Figure 6 (probably you mean "gas to liquid viscosity ratio" $\Pi = \mu_g/\mu$ ?).

- Line 257. According to the common nomenclature, you report the square of the Froude number. It should be $Fr = u_0/\sqrt{ga}$.

- Caption of Figure 7: please, can you define the symbols $R_0$ and $S_0$ ? Moreover, the dashed lines are practically superimposed to the solid lines and difficult to see. Probably you can highlight the dashed lines (eg with thicker lines) or simply indicate in the caption that the MagmaFoam and the Lyakhovsky et al. (1999) solutions practically coincide.

- Line 337. The relationship between maximum volume fraction in the volcanic products and fragmentation was observed by Sparks (1978). Probably it could be appropriate to cite also Sparks (1978) together with La Spina et al., (2017).

**Typos**

- Line 105: "con" $\mapsto$ "can"

- Line 205: "wavelenght" $\mapsto$ "wavelength"

- Line 389: "apporoaches" $\mapsto$ "approaches"

- Line 410: "relativily" $\mapsto$ "relatively"

- Line 411: "theretical" $\mapsto$ "theoretical"

**References**

Sparks, R. S. J.: The dynamics of bubble formation and growth in magmas: A review and analysis, J. Volcanol. Geotherm. Res., 3, 1–37, https://doi.org/10.1016/0377-0273(78)90002-1, 1978.

---

## Referee Comment (RC2)

The paper by Brogi et al. develops a modular framework, MagmaFOAM, for simulating multiphase, multicomponent flow in magmatic systems based on the open-source software package OpenFOAM. The main addition to OpenFOAM are parametrizations of the magma properties, such as density and viscosity, which depend themselves on pressure, temperature and composition. The authors have designed a modular framework that allows users to select and combine model components. I expect that users in the magma-dynamics community will find this approach helpful given that magmatic properties vary dramatically between systems.

Overall, the study is a valuable contribution to the toolboxes available in magma dynamics and I appreciate that the authors provide multiple benchmark computations for their model. However, I think it is important to more clearly explain what this model "is good at" and where its limits lie, particularly since the goal here seems to be to empower a potentially broad community of users to work with models. Upon taking a closer look at the literature (some referenced below), I think the authors will find that the constitutive models they have integrated are more limiting than it might seem at first. In my opinion, an introduction that critically discusses different model approaches, their merits for understanding magma dynamics problems, but also how they relate to work beyond the magma dynamics community strictly defined would be valuable. I have provided some specific suggestions for relevant papers below.

Major concerns:

As the authors point out in the introduction, MagmaFOAM is a mixture model. There is nothing wrong with that. Mixture models have their place and their importance, but particularly since this is a modeling tool that will hopefully be useful for a diverse community of scientists, not all of which think primarily about models, I think it is important to explain very clearly what both the strengths and the limitations of a mixture approximation are. In the paragraph starting on line 33, the authors motivate mixture models as "convenient". I agree that they are, but surely (hopefully?) that is not the primary metric we want to focus on to guide model development.

I suggest that the reviewers rethink and rewrite the paragraph starting on line 33. Currently, it is a very general overview of different modeling techniques, mostly by describing what problems they have been applied to, but less information is provided about the key strengths and weaknesses of different approaches. I think it would be valuable to add that so that readers can make an informed choice about whether this model is useful for what they are trying to understand.

Specific suggestions:
The discussion of direct numerical simulations in the paragraph starting on line 33 could be improved. As the authors well know, the method originated in turbulence research and I would argue that the main claim of fame of this kind of technique is to capture emergent phenomenon in flow. That can be done in the turbulent context and it can be done in a multiphase context, where the long-range hydrodynamic interactions break the symmetry of the flow. The approach can be combined with an interface tracker (and other things), but the

main added value is really to better understand emerging behavior that is difficult to parametrize a-priori.

I think it's misleading to classify Lattice-Boltzmann models (e.g., Huber et al., 2014; Parmigiani et al., 2014) as direct numerical simulations. There is no doubt that Lattice-Boltzmann methods are a valuable approach for mimicking common fluid behavior, particularly in porous media. They are also much less computationally expensive, because they do not solve the Navier-Stokes equation directly and often imply large interface thicknesses. The method itself is completely different from a direct numerical simulation, though.

The authors seem to suggest that mixture models are particularly valuable for small particles and/or high fluid viscosities. The text in its current form seems to suggest that the size of the crystal/bubble/interface determines whether a mixture approach can be adopted or not, but there are several other considerations and there is strong evidence that a mixture approximation is quite problematic in this limit. Over the last two decades, several studies (Segre et al., An effective gravitational temperature for sedimentation, 2001 would be a good starting point to look deeper into that literature) have shown that the behavior of suspensions is particularly complex at low Reynolds number, because interfaces interact over very long distances, leading to surprising emergent behavior. They have shown that these long-range hydrodynamic interactions lead to behavior reminiscent of turbulence even at zero Reynolds number (e.g., Tong et al., Analogies between colloidal sedimentation and turbulent convection at high Prandtl numbers, 1998 etc.). The consequences on the flow field can be dramatic, particularly in the presence of shear (e.g., Qin and Suckale, Flow-to-Sliding Transition in Crystal-Bearing Magma, JGR 2019).

Minor suggestions:
Line 33: I suggest a figure or illustration to convey how drastic the simplification of a multiphase medium through the interpenetrating continuum idea really is to explain this key point to the readers. There is a rich literature on this type of approach with plenty illustrations that they authors might find inspiring.

Line 47 "average forms of the flow equations can be adopted and the need of tracking the exact position of the interface is avoided": Many mixture models (including in this paper) do track interfaces. The most famous example is probably the two-fluid model, which the authors might want to reference for context.

Line 49 "The so-called multi-fluid Eulerian approach": I don't really know what the authors are referring to here. To me, "Eulerian" is a reference system that governing equations can be formulated in (as compared to Lagrangian) rather than an approach. I think it would be valuable to separate the two as many other methods in this paragraph are Eulerian to (e.g., our papers that are cited here, e.g. Suckale et al., 2010a).

Line 55-60: I don't understand which approach/set of governing equations the authors are talking about in this segment. The comment about dispersed phase relaxation is rather generic

to me as is the general issue about computational cost, which I would argue is always a constraint, one way or another. The degree to which relaxation is an issue or not depends on so many things including discretization etc.? And why bring in the pseudo fluid approach and which one are we talking about here specifically? Neither am I convinced that strong thermo-mechanical coupling is the main issue.

Line 98: I agree that the interplay between pressure, temperature, composition and physical processes is the key challenge in modeling volcanic systems. I suggest being more careful with the statement that constitutive models alone can solve the problem, though. Ultimately, constitutive models can only be as good as the equation that they are plugged into, but we do not currently have a continuum equation that applies over the broad range of conditions that volcanic systems traverse with issues arising both in the suspension limit (see the Segre paper I had mentioned above) and in the mush limit, though progress has been made in the context of the mu(I) rheology (e.g., Midi et al., On dense granular flows, 2004; Henann, D. L. & Kamrin, K. A predictive, size-dependent continuum model for dense granular flows, 2013). Needless to say, these complexities would be further amplified by thermal and geochemical effects. Let me emphasize that I do not object to the usage of the constitutive models themselves as that part is unavoidable in a mixture formulation, but with how this path is presented in the text.

Line 164: I appreciate that the authors call out the strong assumptions behind representing bubbles in melt as a monodisperse periodic array of static spheres, but I do not think that the monodisperse size distribution is necessarily the main crime here. Bubbles are not static, not even when they are so small that they do not move very fast themselves, because of the long-range hydrodynamic interactions connecting them and leading to self-organization, as manifested in bubble waves (e.g., Manga, Waves of bubbles in basaltic magmas and lavas, JGR, 1996). I have no problem with this component being integrated into the model, but I do not think that the claim that it represents "an accurate representation of the coupled momentum balance and diffusive transport of volatiles" is warranted. Similarly, I'm not convinced that the method produces "accurate results especially at low vesicularity". That is a rather strong statement. I'd be happy to be convinced if similarly strong evidence is provided to back this up.

Line 180: The trick with these interface tracking techniques is of course what to do with the mass enclosed in an interface that drops below the grid resolution. The momentum equation is no longer off help in that case, because flow is not resolved at the subgrid scale. So yes, VOF methods are generally conservative, because they redistribute the subgrid mass, but significant error in the interface position can arise from that approach (I am guessing that is what the authors mean by "numerical blur"). I like the term "numerical blur", but in the interest of enabling users to understand the capabilities of this software as much as possible, I think it's worth not only mentioning it, but actually explaining where it comes from. In addition to the blur aspect, thought, it's also worth keeping in mind that distortions to the interface can build up, leading to seemingly sharp interface features, similar to particle-tracking of interfaces or marker chains, e.g., Van Keken et al., A comparison of methods for the modeling of thermochemical convection, JGR, 1997). I think it's worth adding a bit more explanation of the method, how it conserves mass, and what the potential drawbacks of that approach are.

Line 188: I think it would be useful for the authors to refer to an actual figure or test case here, before concluding that they find "remarkably good agreement". That would make it easier for the reader to assess whether they are convinced of the statement. I do realize that the testcases are presented in the next sections, but it's a bit odd to present the conclusion prior to showing the benchmark results.

Line 196: I entirely agree that the Rayleigh-Taylor instability is a great benchmark for fluid solvers, but I am not sure that I would present it as a benchmark of "magma mixing". The specific growth rate referred to in this section assumes two immiscible fluids, and only holds strictly in that specific limit. To me, it's a touch odd to describe the overturn dynamics of two immiscible fluids as mixing.

Line 233: Are these melts assumed to be immiscible or miscible? In other words, are they separated by a sharp interface or is there a compositional field variable that may start as sharp but can diffuse over time? Not entirely clear to me.

Line 260: I would be careful with the statement that "Reynolds number mainly controls bubble stability and breakup". There is no doubt that Reynolds is very important here, because the stagnation pressure at finite Re strongly deforms the bubble and deformation will be further amplified when turbulence kicks in. My concern with the statement is that a cursory reader could interpret this as "bubbles at low Reynolds number do not break up". That's obviously not true and I do not think that the authors want to insinuate that (as their later statement clarifies). The explanation provided at the end of the paragraph (based on Eo and Re) is much more clear.

Line 273: There is an issue in the typesetting here (line break needs removing).

Line 395: I struggle with this last paragraph. The authors make big promises here, e.g., "the inclusion of Lagrangian tracers will result in a more detailed description of the micro-physics", but do not offer a lot of evidence to back up this claim. Yes, population balance equation and Lagrangian tracers are convenient, but also have many drawbacks and it is not clear to me how they specifically advance our understanding of the micro-physics as I think of them as limited that way (after all, the "micro-physics" is largely thrown out of these very approaches). For these reasons, this last paragraph strikes me as rather speculative and a bit vague.

Overall, I think MagmaFOAM is a valuable contribution to the models available in the volcanological community. I hope my comments are helpful and I would be happy to clarify and/or discuss these suggestions if the authors want.

Jenny Suckale

---

## Author Response (AR1)

**Response to Reviewer #1's comments to: "MagmaFOAM-1.0: a modular framework for the simulation of magmatic systems" by Brogi et al.**

**Manuscript submitted for publication on GMD**

**Description**

This work describes a software library, which extends OpenFOAM, dedicated to the solution of problems typically encountered in modeling volcanic processes.

**General comments**

The library contains already existing subroutines (eg: SOLWCAD) and new interesting software. However, the existing software has been recoded within the same modular framework (OpenFOAM), allowing an easier use by model developer. I have appreciated the efforts of the authors and I think that MagmaFOAM will be very useful for the volcanological community.

The authors gratefully appreciate the time and effort the referee has dedicated to provide a constructive review of the manuscript. The positive feedback pushes us to look forward with enthusiasm to future developments of MagmaFOAM that may be of interest for the volcanological community.

Below, we report the point-by-point answer to the reviewer's minor comments.

Federico Brogi, on behalf of all authors

**Minor comments**

Line numbers referenced below by us refer to the revised version of the manuscript.

- Of course, eq.(3) is valid for T > C

Following the reviewer's suggestion, we added in the text (line 127) that eq. 3 requires $A+B/(T-C)>0$. This condition in fact implies $T>C$ as a necessary condition

knowing that A is negative and B positive (see fig. 4 in Giordano et al., 2008). However, T>C requires only B/(T-C)>0 instead of B/(T-C)>-A.

line 127: *"Let us also note that eq. 3 is valid only for A+B/(T-C)>0."*

- Line 256: The Reynolds number is more often based on the bubble diameter that bubble radius.

We completely agree with the reviewer. However, we prefer to use the Reynolds number definition of Suckale et al., 2010a (with the bubble radius) in order to avoid confusion when comparing our results with the ones reported by these authors. For sake of clarity, we also added a note in the text. (line 259)

line 259: *"Let us note that to be consistent with Suckale et al. (2010a) all non-dimensional numbers here are based on the bubble radius instead of the bubble diameter, which is also commonly used in the literature (e.g. Roghair et al. 2011))."*

- Line 257: The definition of the Weber number seems incorrect. It should contain the surface tensionin the denominator. Please check.

We thank the reviewer, indeed the definition of the Weber number is wrong. It has been replaced with the correct one containing the surface tension.

line 257: *"... Weber Number $We = \rho v_0^2 a / \sigma$ "*

- The definition of $\Pi = \mu/\mu_g$ at line 258 seems incompatible with its value ($10-6$) reported at line 259 and in the caption of Figure 6 (probably you mean "gas to liquid viscosity ratio" $\Pi = \mu_g/\mu$ ?).

We thank the reviewer, it is in fact the "gas to liquid viscosity ratio". The definition has now been corrected.

line 258: *"... and gas to liquid viscosity ratio $\Pi = \mu_g/\mu$. "*

- Line 257. According to the common nomenclature, you report the square of the Froude number. It should be Fr = $u_0/\sqrt{ga}$.

We agree with the reviewer. We corrected the Froude number definition (line 257) and added a comment in the caption of Figure 6 for clarity. As for the Reynolds

number we use the square of Fr to be consistent with the reference study (Suckale et al., 2010a).

line 257: "*... Froude number* $Fr \; = \; v_0/\sqrt{ga}$"

Caption of Figure 6: "*Simulation of bubble rise in a basaltic melt using* `interFoam` *are compared with the results of Suckale et al. (2010a) (blacklines) for three different regimes: (a) No breakup (Re≈5, $Fr^2$≈0.4, We≈90, and Π = 10−6), (b) Gradual breakup (Re≈25, $Fr^2$≈0.3; We≈800 and Π=10−6); (c) Catastrophic breakup (Re≈250, $Fr^2$≈0.16, We≈1350 and Π = 10−6). For each regime, snapshots at different non-dimensional times are shown. To be consistent with Suckale et al. (2010a) here we use the square of the Froude number ($Fr^2 = v_0^2/ga$).*"

- Caption of Figure 7: please, can you define the symbols $R_0$ and $S_0$ ? Moreover, the dashed lines are practically superimposed to the solid lines and difficult to see. Probably you can highlight the dashed lines (eg with thicker lines) or simply indicate in the caption that the MagmaFoam and the Lyakhovsky et al. (1999) solutions practically coincide.

We have modified the caption according to the reviewer's suggestion.

Caption of Figure 7: "*Temporal evolution of bubble radius for an instantaneous decompression from $p_0$=150 MPa to $p_L$= 120 MPa. In blue the comparison between the MagmaFOAM model* `multiComponentODERPShellDStatic` *(solid lines) and numerical solutions from Lyakhovsky et al. (1996) (dashed lines) that practically coincide for three different values of diffusion coefficient of $H_2O$. The red lines represent the same simulations with 1 wt% of $CO_2$ added in the melt. The diffusion coefficient of $CO_2$ is one order of magnitude smaller than $H_2O$. Initial conditions and parameters (see Appendix B) in all simulations are: $\rho_L$= 2300 kg/m³, μ = 5·10⁴ Pa s, σ=0.06 N/m, T=1123 K, $p_G$(t=0)=$p_0$+2σ/R(t=0), R(t=0)=10⁻⁷ m, $S_0$=2·10⁻⁴ m, $C^0_{H2O}$= 5.3 wt%. Saturation concentration is computed using SOLWCAD (Papale et al., 2006).*"

- Line 337. The relationship between maximum volume fraction in the volcanic products and fragmentation was observed by Sparks (1978). Probably it could be appropriate to cite also Sparks (1978) together with La Spina et al., (2017).

Following the reviewer's suggestion, we added the missing reference (line 339). Moreover, we also consistently modified the notation used in the bubble growth model (Appendix B) and the model for volatile concentration at the bubble-melt interface (section 2.2).

line 339: "*... using a critical volume fraction criterion (0.5< α <0.7; e.g. Sparks 1978 or La Spina et al. 2017).*"

Reference: Sparks R. S. J. (1978) The dynamics of bubble formation and growth in magmas: a review and analysis. Journal of Volcanology and Geothermal Research 3:1–37.

**Typos**

- *Line 105: "con" → "can"*

- *Line 205: "wavelenght" → "wavelength"*

- *Line 389: "apporoaches" → "approaches"*

- *Line 410: "relativily" → "relatively"*

- *Line 411: "theretical" → "theoretical"*

All signalled typos have been corrected, thanks again to the reviewer for noticing them.

**Response to Reviewer #2's comments to: "MagmaFOAM-1.0: a modular framework for the simulation of magmatic systems" by Brogi et al.**

**Manuscript submitted for publication on GMD**

The paper by Brogi et al. develops a modular framework, MagmaFOAM, for simulating multiphase, multicomponent flow in magmatic systems based on the open-source software package OpenFOAM. The main addition to OpenFOAM are parametrizations of the magma properties, such as density and viscosity, which depend themselves on pressure, temperature and composition. The authors have designed a modular framework that allows users to select and combine model components. I expect that users in the magma-dynamics community will find this approach helpful given that magmatic properties vary dramatically between systems.

Overall, the study is a valuable contribution to the toolboxes available in magma dynamics and I appreciate that the authors provide multiple benchmark computations for their model. However, I think it is important to more clearly explain what this model "is good at" and where its limits lie, particularly since the goal here seems to be to empower a potentially broad community of users to work with models. Upon taking a closer look at the literature (some referenced below), I think the authors will find that the constitutive models they have integrated are more limiting than it might seem at first. In my opinion, an introduction that critically discusses different model approaches, their merits for understanding magma dynamics problems, but also how they relate to work beyond the magma dynamics community strictly defined would be valuable. I have provided some specific suggestions for relevant papers below.

The authors thank Prof. Jenny Suckale for a thorough and insightful review of the paper that provided us with a valuable different perspective on our study. We have modified the manuscript trying to incorporate all the suggestions, with the hope that it is now more accurate, readable and, more importantly, easily accessible to a broader community of potential users. Specifically, we have tried to expand the manuscript to include more comments on the validity and applicability of different approaches to solve CFD problems, both within the magma dynamics community and beyond. We believe the manuscript has improved significantly thanks to the reviewer's inputs and we hope it is now suitable for publication in GMD.

Below, we report the point-by-point answer to the reviewer's comments.

Federico Brogi, on behalf of all authors

Line numbers referenced below by us refer to the revised version of the manuscript.

Major concerns:

As the authors point out in the introduction, MagmaFOAM is a mixture model. There is nothing wrong with that. Mixture models have their place and their importance, but particularly since this is a modeling tool that will hopefully be useful for a diverse community of scientists, not all of which think primarily about models, I think it is important to explain very clearly what both the strengths and the limitations of a mixture approximation are. In the paragraph starting on line 33, the authors motivate mixture models as "convenient". I agree that they are, but surely (hopefully?) that is not the primary metric we want to focus on to guide model development.

I suggest that the reviewers rethink and rewrite the paragraph starting on line 33. Currently, it is a very general overview of different modeling techniques, mostly by describing what problems they have been applied to, but less information is provided about the key strengths and weaknesses of different approaches. I think it would be valuable to add that so that readers can make an informed choice about whether this model is useful for what they are trying to understand.

We thank the reviewer for this comment, which sparked a re-thinking and re-writing of the whole manuscript Introduction. We expanded the discussion on interface-resolving and mixture models (single or multi-fluid) in order to help the reader to better understand their strengths and weaknesses. As the reviewer is well aware of, a complete review of the validity of different mixture modeling approaches (multi-fluid or single fluid) for every application is out of the scope of this work. Nevertheless, we have tried to include a first-order evaluation of pros and cons of different fluid modeling approaches.

Specific suggestions:

The discussion of direct numerical simulations in the paragraph starting on line 33 could be improved. As the authors well know, the method originated in turbulence research and I would argue that the main claim of fame of this kind of technique is to capture emergent phenomenon in flow. That can be done in the turbulent context and it can be done in a multiphase context, where the long-range hydrodynamic interactions break the symmetry of the flow. The approach can be combined with an interface tracker (and other things), but the main added value is really to better understand emerging behavior that is difficult to parametrize a-priori.

The paragraph starting on line 33 has been largely reshaped. We have addressed the reviewer's concerns by including a much longer discussion on the advantages and limitations of different modeling approaches to tackle different kinds of problems (lines 33-63).

I think it's misleading to classify Lattice-Boltzmann models (e.g., Huber et al., 2014; Parmigiani et al., 2014) as direct numerical simulations. There is no doubt that Lattice-Boltzmann methods are a valuable approach for mimicking common fluid behavior, particularly in porous media. They are also much less computationally expensive, because they do not solve the Navier-Stokes equation directly and often imply large interface thicknesses. The method itself is completely different from a direct numerical simulation, though.

Similar to DNS, Lattice Boltzmann method can resolve all the scales of the flow and the interface to study complex interfacial dynamics and interactions. The two cited references are examples of applications to volcanic system modelling. We have slightly modified the wording in the paragraph to avoid misunderstandings, following the reviewer's suggestion:

line 59: *Based on the computationally efficient Lattice Boltzmann method, interface resolving modeling has been also useful to better understand bubble growth, deformation and coalescence (Huber et al., 2014) as well as the mush microphysics characterizing crystal-rich magma reservoirs (Parmigiani et al., 2014).*

The authors seem to suggest that mixture models are particularly valuable for small particles and/or high fluid viscosities. The text in its current form seems to suggest that the size of the crystal/bubble/interface determines whether a mixture approach can be adopted or not, but there are several other considerations and there is strong evidence that a mixture approximation is quite problematic in this limit. Over the last two decades, several studies (Segre et al., An effective gravitational temperature for sedimentation, 2001 would be a good starting point to look deeper into that literature) have shown that the behavior of suspensions is particularly complex at low Reynolds number, because interfaces interact over very long distances, leading to surprising emergent behavior. They have shown that these long-range hydrodynamic interactions lead to behavior reminiscent of turbulence even at zero Reynolds number (e.g., Tong et al., Analogies between colloidal sedimentation and turbulent convection at high Prandtl numbers, 1998 etc.). The consequences on the flow field can be dramatic, particularly in the presence of shear (e.g., Qin and Suckale, Flow-to-Sliding Transition in Crystal-Bearing Magma, JGR 2019).

The authors thank the reviewer for highlighting this missing aspect that is now considered in the introduction of the paper. We have also expanded and better explained the paragraph regarding the computational issue of multi fluid models with small relaxation times (small particles/high viscosity; lines 86-93). This is also demonstrated later on in the paper with the shock tube test (lines 420-433). In the introduction we also clarified how with a single fluid mixture this issue can be effectively removed (lines 93-99).

Minor suggestions:

Line 33: I suggest a figure or illustration to convey how drastic the simplification of a multiphase medium through the interpenetrating continuum idea really is to explain this key point to the readers. There is a rich literature on this type of approach with plenty illustrations that they authors might find inspiring.

As the reviewer suggests, we have added two illustrations (Figure 1) in order to help the reader understand the multiscale character of multiphase flows as well as the simplification of considering a dispersed phase made of discrete elements as continuum fluid.

Line 47 "average forms of the flow equations can be adopted and the need of tracking the exact position of the interface is avoided": Many mixture models (including in this paper) do track interfaces. The most famous example is probably the two-fluid model, which the authors might want to reference for context.

We agree with the reviewer that this statement is ambiguous and can be misleading. We have in fact removed this sentence and included in the introduction a more in-depth discussion of the mixture approximation, better highlighting its strengths and limitations (lines 64-109).

In particular, regarding the issue mentioned above

line 68: *The multi fluid formulation employs averaging techniques that filter out the interfacial scales that are too small to be resolved [Marschall et al., 2013]. The complexity of a volume with multiple phases at the local scale is characterised by phase-average properties and a volumetric fraction that expresses the relative presence of one phase with respect to the others (Figure 1b). Neglecting the details of the topology of the interfaces at the local scale allows to describe the phases at the system scale as interpenetrating continua governed by separate sets of conservation equations. The resulting equations hence resemble those for single phase flows except for the volumetric fraction and the presence of phase interaction terms that require appropriate closure. Similarly to Large Eddy Simulations for turbulent flows, additional constitutive models are in fact required to recover the physics of the missing small scales.*

Line 49 "The so-called multi-fluid Eulerian approach": I don't really know what the authors are referring to here. To me, "Eulerian" is a reference system that governing equations can be formulated in (as compared to Lagrangian) rather than an approach. I think it would be valuable to separate the two as many other methods in

this paragraph are Eulerian to (e.g., our papers that are cited here, e.g. Suckale et al., 2010a).

We agree with the reviewer that Eulerien and Lagrangian are reference systems in which conservation equations can be formulated in. The term 'Eularian' here is not necessary and may only add an unnecessary complexity for the reader; therefore it has been removed.
In the multiphase literature and multiphase CFD (e.g. OpenFOAM) the term "Eulerian" is commonly used (also in textbooks, e.g. Yeoh and Tu, 2019: Computational Techniques For Multiphase Flows) to refer to the continuous (average/mixture) field phase modeling as opposed to discrete particle phase modeling (Lagrangian). For instance, Eulerian-Eulerian is used to refer to two/multi-fluid models whereas Euelerian-Lagrangian refers to a hybrid approach (continuous phase + discrete particles model).

Line 55-60: I don't understand which approach/set of governing equations the authors are talking about in this segment. The comment about dispersed phase relaxation is rather generic to me as is the general issue about computational cost, which I would argue is always a constraint, one way or another. The degree to which relaxation is an issue or not depends on so many things including discretization etc.? And why bring in the pseudo fluid approach and which one are we talking about here specifically? Neither am I convinced that strong thermo-mechanical coupling is the main issue.

We have now specified that we are referring to the multi-fluid equations and better explained the meaning of the relaxation time. In particular, we explicitly refer to the fact that relaxation is introduced by the definition of the constitutive models for the phase coupling terms (e.g fluid-particle drag).

line 86-98: *Multi fluid models are, however, more computationally expensive than single phase models, as they require an additional set of governing equations for each phase. As the number of phases increases, the computational burden also increases dramatically (e.g. Ferry and Balachandar, 2001). The definition of the interfacial exchange terms can also indirectly increase the computational cost. For instance, the fluid-particle drag introduces a time scale in the equations, the relaxation time of the dispersed phase, that describes the time required by the particle to adapt to a change in velocity of the surrounding fluid. When this relaxation time is small (typically for small particles and/or high fluid viscosities), the stability and accuracy of the numerical solution require a time step smaller than the relaxation time, increasing the number of iterations needed to solve the flow time scale. Under the assumption of thermo-mechanical equilibrium the equations of the multi fluid model can be further reduced to an evolution equation for a single pseudo-fluid representing a mixture of multiple phases. From a computational point of view, given the reduced number of equations needed to track the evolution of the*

*mixture, this is a more convenient approach. In addition, when there is a strong thermo-mechanical coupling between phases (small relaxation times), it is reasonable to assume that the particle velocity is equal to the fluid velocity, effectively removing the aforementioned issues related to the definition of the interaction terms and the relaxation time.*

We believe the relaxation time issue is a crucial aspect that is normally not well highlighted in the literature on multi-fluid models, even if it will invariably arise when using these models. The drag term (fluid-particle interaction) introduces a timescale that describes the time the particle needs to re-equilibrate to a change in fluid velocity. For individual point particles (using Stokes' law [e.g. Cerminara et al.,2016]) this time is:

(lines 343-350) "... *proportional to the square of the bubble diameter and inversely proportional to the kinematic viscosity of the continuous liquid phase ... In magmatic phenomena, when considering small bubbles (e.g.,100µm) and even relatively low viscosities (e.g., 10 Pa s) tau can reach very small values (10^-6 s), resulting in very strong mechanical phase coupling.*"

So the relaxation time sets a limit on the deltaT to be used in simulations that is much smaller than the time scale of the flow phenomena at the much larger spatial scale for which multi-fluid solver are used. The great disparity between the two time scales (relaxation and integral flow time scale) makes the multi-fluid simulations computationally too expensive. In the pseudo/single-fluid model instead one may neglect the phase coupling term (reasonably assuming that the particle velocity is equal to the fluid velocity) and the relaxation time is removed from the equations.

Line 98: I agree that the interplay between pressure, temperature, composition and physical processes is the key challenge in modeling volcanic systems. I suggest being more careful with the statement that constitutive models alone can solve the problem, though. Ultimately, constitutive models can only be as good as the equation that they are plugged into, but we do not currently have a continuum equation that applies over the broad range of conditions that volcanic systems traverse with issues arising both in the suspension limit (see the Segre paper I had mentioned above) and in the mush limit, though progress has been made in the context of the mu(I) rheology (e.g., Midi et al., On dense granular flows, 2004; Henann, D. L. & Kamrin, K. A predictive, size-dependent continuum model for dense granular flows, 2013). Needless to say, these complexities would be further amplified by thermal and geochemical effects. Let me emphasize that I do not object to the usage of the constitutive models themselves as that part is unavoidable in a mixture formulation, but with how this path is presented in the text.

The limits of the continuum phase approximation, together with the need for constitutive equations, that have their own limits, are now thoroughly discussed in the introduction of the paper. Here, we have specified that these constitutive models represent the basic necessary ingredients to deal with this complexity:

line 145: *When handling this thermo-physical complexity, state-of-the-art multi-component constitutive models that compute melt properties as a function of the local pressure, temperature and composition are the necessary basic ingredients and have been implemented in MagmaFOAM.*

Line 164: I appreciate that the authors call out the strong assumptions behind representing bubbles in melt as a monodisperse periodic array of static spheres, but I do not think that the monodisperse size distribution is necessarily the main crime here. Bubbles are not static, not even when they are so small that they do not move very fast themselves, because of the long-range hydrodynamic interactions connecting them and leading to self-organization, as manifested in bubble waves (e.g., Manga, Waves of bubbles in basaltic magmas and lavas, JGR,1996). I have no problem with this component being integrated into the model, but I do not think that the claim that it represents "an accurate representation of the coupled momentum balance and diffusive transport of volatiles" is warranted. Similarly, I'm not convinced that the method produces "accurate results especially at low vesicularity". That is a rather strong statement. I'd be happy to be convinced if similarly strong evidence is provided to back this up.

We agree with the reviewer that our sentence in the text is misleading. The momentum balance to which we are referring here is the Rayleigh-Plesset (R-P) equation, that includes only the effects related to bubble growth for a single, static and spherical bubble. Solving the Rayleigh-Plesset equation, in combination with the advection diffusion equation, the diffusion profile in the melt shell surrounding the bubble can be accurately resolved to provide the mass flux toward the bubble. Compared to other approaches (see for example Huber et al. 2014), this method is not affected by numerical diffusion when solving the gas-melt interface; at the cost of an ideal geometry. When dealing with low vesicularities, the monodisperse assumption seems to hold relatively well, as it is confirmed by experiments (Coumans et al., 2020). Mechanical decoupling is not taken into account by the model, hence this approach is valid only when the time scale of the process under study is much shorter than the mechanical relaxation time.

The text has been modified as follows:

line 211: *This approach provides, at low computational cost, an accurate representation of the coupled momentum balance and diffusive transport of volatiles, because it well resolves the concentration profile near the bubble interface (Huber et al., 2014). The strong assumptions that the size distribution is monodisperse and the bubbles are non deformable and mechanically coupled with melt, limits the range of applicability of the model. In high-viscosity systems at low vesicularity, the model can provide reliable results when compared with experiments (Coumans et al, 2020). The model does not take into account interfacial interactions (fluid-particle and*

*particle-particle) that can give rise to emergent behaviour, as in the case for example of bubble waves (Manga, 1996).*

Line 180: The trick with these interface tracking techniques is of course what to do with the mass enclosed in an interface that drops below the grid resolution. The momentum equation is no longer of help in that case, because flow is not resolved at the subgrid scale. So yes, VOF methods are generally conservative, because they redistribute the subgrid mass, but significant error in the interface position can arise from that approach (I am guessing that is what the authors mean by "numerical blur"). I like the term "numerical blur", but in the interest of enabling users to understand the capabilities of this software as much as possible, I think it's worth not only mentioning it, but actually explaining where it comes from. In addition to the blur aspect, thought, it's also worth keeping in mind that distortions to the interface can build up, leading to seemingly sharp interface features, similar to particle-tracking of interfaces or marker chains, e.g., Van Keken et al., A comparison of methods for the modeling of thermochemical convection, JGR, 1997). I think it's worth adding a bit more explanation of the method, how it conserves mass, and what the potential drawbacks of that approach are.

Following the reviewer suggestion, we have significantly expanded this paragraph in order to better explain VOF mass conservative properties and the drawbacks of this methods:

line 224: *The Volume of Fluid method (VOF) is adopted in OpenFOAM to resolve the position and shape of the interface separating  two fluids or phases (e.g. liquid-gas). This methodology treats the interface discontinuity as a smooth but rapid variation (few computational cells) of an indicator field (volumetric fraction) representing the relative presence of one phase with respect to the other in each cell. The volumetric fraction is zero or one away from the interface, allowing to distinguish between one phase and the other, and assumes intermediate values in the region containing the interface. As a result, the location of the interface and its shape are known only implicitly from the volumetric fraction. The evolution of the interface is then obtained by simply advecting the volumetric fraction using the velocity field computed from a single (e.g. the OpenFOAM solver `interFoam`) or multi-fluid momentum equation (e.g. the OpenFOAM solver `multiphaseEulerFoam`).The transport equation for the indicator function is under the constraint of mass conservation and therefore, with respect to other methods (e.g. Level-set method), VOF is mass conservative by construction. However, in practice the conservation of mass depends on the accuracy in solving numerically the transport equation. The discontinuous nature of the volumetric fraction (a step function) at the interface makes the numerical solution of this equation challenging. In particular, numerical diffusion due to the discretization of the advection term prevents a sharp representation of the interface that tends to*

*be smeared over the computational cells causing inaccurate estimations of its position and curvature. Different techniques exist  to solve this issue. With a geometrical approach one may reconstruct the position of the discontinuity at the subgrid level, provided that the interface can be described with a specific functional form (Rider and Kothe, 1998, Aulisa et al., 2003). The interface is then advected by the flow in a lagrangian manner. This technique effectively prevents numerical diffusion and provides a more accurate representation of the interface at the cost of a significantly  more complex algorithm and increased computational load. Other approaches rely on relatively more simple algebraic solutions that reduce  numerical diffusion (e.g. Ubbink and Issa, 1999).*

*Specifically, `interFoam`  makes use of a high order differencing scheme (in the interface region only) and an additional compressive term in the advection equation that effectively counterbalances the numerical diffusion of the interface. While this approach is simpler and less computationally expensive than the geometrical reconstruction, the interface is spread over few  computational cells and its precise position remains unknown. Nevertheless, in kinematic tests, interFoam has shown good mass conservation properties and acceptable advection errors (Deshpande et al., 2012).*

*Spurious currents and artificial deformations of the interface are also an issue with VOF. Inaccurate interface curvature, together with a discrete force imbalance at the interface, typically produce spurious vortices that can artificially deform the interface. Depending on the simulation setup, the kinetic energy of these vortices may rapidly decay or grow and in the worst case scenario even cause the simulation to crash.*

*However, spurious currents may pose a serious issue mostly for surface tension dominated flows and are less important for inertia dominated flows. For interFoam, Deshpande et al., (2012) have shown that the growth of spurious currents can be controlled by choosing appropriate time steps.*

Line 188: I think it would be useful for the authors to refer to an actual figure or test case here, before concluding that they find "remarkably good agreement". That would make it easier for the reader to assess whether they are convinced of the statement. I do realize that the test cases are presented in the next sections, but it's a bit odd to present the conclusion prior to showing the benchmark results.

As suggested by the reviewer, we have added some references to the figures in the following sections. We believe that a short summary of the results may help the reader to know the main outcomes of the benchmark computations described in more detail in the text below.

line 258: *Overall, we find a remarkably good agreement between our simulation results and theoretical or numerical results from literature, over different flow regimes of interest for magma dynamics. The numerical solutions relative to cases with low Reynolds number Re are very accurate (e.g. Figure 4 and 7a). At larger Re, the*

*results are less accurate due to the appearance of high frequency numerical noise that can trigger secondary spurious interface instabilities (e.g. Figure 5). Reducing numerical noise by adopting different numerical schemes is one relevant element for future investigation.*

Line 196: I entirely agree that the Rayleigh-Taylor instability is a great benchmark for fluid solvers, but I am not sure that I would present it as a benchmark of "magma mixing". The specific growth rate referred to in this section assumes two immiscible fluids, and only holds strictly in that specific limit. To me, it's a touch odd to describe the overturn dynamics of two immiscible fluids as mixing.

The reviewer is right, the use of 'mixing' was a bit far-stretched in this paragraph. We have modified the title, and added a more complete explanation of the relevance of Rayleigh-Taylor instabilities for magmatic dynamics.

line 267: *Magma is thought to rise from the mantle into the crust in discrete batches (Annen et al., 2006) that then tend to stall and cool at different depths, while their chemistry evolves towards more felsic compositions (Sigurdsson, 2015). Different batches of magma may interact as they ascend towards shallower depths, resulting in magma mingling and mixing. The latter are widespread phenomena in volcanic plumbing systems (Perugini et al., 2012; Morgavi et al., 2017) and have often been invoked as eruption triggers (Wark et al., 2007; Druitt et al., 2012; Martì et al., 2020). Mingling and mixing are typically driven either by gravitational Rayleigh-Taylor instabilities, involving contacts between magmas with different densities due to compositional, thermal or phase stratifications (e.g., Jellinek et al., 1999; Montagna et al., 2015; Garg et al., 2019); or by percolation of pressurized magmas arriving from depth into mushy reservoirs (Bachmann et al., 2003; Seropian et al., 2018).*

[...]

line 306: *Magmas usually interact both mechanically and chemically, therefore the immiscible approximation described above is not justified a-priori. Nevertheless, to first approximation and on relatively short time scales (hours to days), chemical diffusion among interacting magmas at the plumbing system scale can be neglected (e.g., Ruprecht et al., 2008; Garg et al., 2019), and magmas can be considered immiscible. Here we describe exemplary buoyancy-driven interaction among two natural silicate melts (Figure 6).*

Line 233: Are these melts assumed to be immiscible or miscible? In other words, are they separated by a sharp interface or is there a compositional field variable that may start as sharp but can diffuse over time? Not entirely clear to me.

We have reinstated in the text that the melts are assumed to be immiscible to clarify this point (no compositional variable is considered in the simulation). See also the answer above.

line 313: *Melt compositions are reported in Table D1. Here, the composition and p, T conditions are considered only in the pre-processing to compute the density and viscosity of the melt that remain constant throughout the simulation. The relevant dimensionless numbers are now ...*

Line 260: I would be careful with the statement that "Reynolds number mainly controls bubble stability and breakup". There is no doubt that Reynolds is very important here, because the stagnation pressure at finite Re strongly deforms the bubble and deformation will be further amplified when turbulence kicks in. My concern with the statement is that a cursory reader could interpret this as "bubbles at low Reynolds number do not break up". That's obviously not true and I do not think that the authors want to insinuate that (as their later statement clarifies). The explanation provided at the end of the paragraph (based on Eo and Re) is much more clear.

We agree with the reviewer that this sentence may be misleading and so we decided to change it.

line 340: *The Reynolds and Eotvos numbers control bubble stability, deformation and breakup. Indicatively, for Eo < 1 and Re < 1 the bubble is stable and preserves its initial spherical shape even in the presence of small perturbations of its interface. For Eo > 1 and Re < 1 the bubble deforms and may breakup if random ...*

Line 273: There is an issue in the typesetting here (line break needs removing).

We thank the reviewer for noticing this issue, the line break is now removed.(line 352)

Line 395: I struggle with this last paragraph. The authors make big promises here, e.g., "the inclusion of Lagrangian tracers will result in a more detailed description of the micro-physics", but do not offer a lot of evidence to back up this claim. Yes, population balance equation and Lagrangian tracers are convenient, but also have many drawbacks and it is not clear to me how they specifically advance our understanding of the micro-physics as I think of them as limited that way (after all, the "micro-physics" is largely thrown out of these very approaches). For these reasons, this last paragraph strikes me as rather speculative and a bit vague.

This paragraph has now been considerably expanded to really highlight the added value and the limitations of both lagrangian and population balance modeling.

*line 473: The tool is meant to be under continuous development, already underway. The addition of population balance equations to single and multi fluid models to statistically describe the dispersed phases (bubbles and crystals, Marchisio and Fox, 2013) will improve our understanding of how polydispersity can impact magmatic system evolution (Colucci et al., 2017a; de' Michieli Vitturi and Pardini, 2020).*

*In large-scale multi-fluid simulations, the exchanges of mass, momentum, and energy through the interface between phases need to be modelled accurately to determine the rate of phase change and the degree of mechanical and thermal disequilibrium between phases. The population balance is a statistical approach for modelling the mesoscale dynamics, widely used in chemical engineering, which describes the temporal and space evolution of a large number of particles through a number distribution function (Yeoh and Tu, 2019). In this way microscopic processes involving bubble dynamics and interactions between bubbles can be included in large-scale multifluid simulations. In fact, DNS allows to model particle-particle interactions and capture emerging behaviours in complex systems; however, the large quantity of microphysics taken into account in DNS has to be filtered and condensed in a sub-model to be used in large-scale simulations. Mesoscopic models represent intermediate models that describe, through a set of mesoscale variables, the microphysics of the system. The formulation of population balance requires adequate closure models for the microphysics that can be developed with the aid of experimental (Mancini et al., 2016) and DNS investigations (Marchisio and Fox, 2013).*

*The inclusion of Lagrangian tracers will result in a more detailed description, with respect to multi fluid models, of the micro-physics that determines the macroscopic properties driving the dynamics. In the Eulerian-Lagrangian approach, bubbles are treated as discrete Lagrangian particles in an ambient Eulerian continuous flow. (e.g., Ghahramani et al., 2019). This approach in fact is more appropriate than multi fluid models when the number of particles is too small to be treated as a continuum, or when single particles' behaviour (e.g. rapidly expanding/contracting bubbles) is so specific that they are not well represented by unique averaged fields density, velocity or temperature (e.g. Ghahramani et al., 2019). With respect to the DNS approach, where bubble-bubble and bubble-melt interactions emerge self-consistently, in the Eulerian-Lagrangian models phase interactions are defined by constitutive models. However, the Eulerian-Lagrangian approach, compared to the DNS, allows to simulate larger populations of particles at a much lower computational cost. The study of complex mixing behaviour in magma mushes is only an example of possible applications (Bergantz et al., 2015).*

Overall, I think MagmaFOAM is a valuable contribution to the models available in the volcanological community. I hope my comments are helpful and I would be happy to clarify and/or discuss these suggestions if the authors want.

Jenny Suckale

We again thank Prof. Jenny Suckale for the time and effort has dedicated to review the paper. Her contribution has been valuable to improve our manuscript and make it truly accessible to the larger volcanological community.

Federico Brogi, on behalf of all authors

---

## Author Response (AR2)

The reviewer's reports have come in and I am happy to inform you that your manuscript has been accepted subject to minor revision. Could you please update your manuscript to address the comments and suggestions by reviewer #1.
Thanks and all the best.

Lutz Gross

The authors thank the editor and the reviewers for the time and effort they have dedicated to provide a constructive revision of this work. All comments have been of great help to improve our paper. The manuscript is now updated to address the latest technical comments from reviewer #1, and we believe that it is now suitable for publication in GMD.

Below, we report the point-by-point answer to the reviewer's minor comments.

Federico Brogi, on behalf of all authors

**Referee #1**

Line 174: I agree that, in general, the viscosity of magma expressed in Pas, is a number greater than 1 so that $\log \eta > 0$. However, I thing that the range of validity of eq.(3) is more affected by the behaviour at low temperatures, when $T$ reaches the value of $C$ and the expression on the right diverges. It should be noted that the right hand of eq.(3) has no physical meaning for $T \le C$, not for $\log \eta < 0$.

Following the reviewer's suggestion, we have now clearly stated in the paper that eq. (3) is not valid for $T \le C$ and added a relevant reference.

line 174: *Let us also note that eq. 3 has no physical meaning for T ≤C (Mauro et al., 2009).*

Equation (4): The use of the diffusion coefficient $D$ (in terms of gradient of concentration) and of the mass transfer coefficient $k$ (in terms of difference of concentration between two points) are alternatives. Please, check if both $k$ and $D$ are required in your transport equation. You can also easily perform dimensional analysis to check that.

We thank the reviewer for spotting this: Equation (4) is indeed not correct. Following the reviewer input we have expanded and corrected this paragraph to make it clearer and consistent. In order to avoid ambiguous or misleading definitions, the units for each quantity we refer to in the equations are now specified.

line 176: *Models accounting for multicomponent phase change require a description of the evolution of the composition at the interface between phases. The mass transfer rate (per unit volume of liquid+gas) of a volatile component can be defined as the product between the interfacial mass flux $J_i$ [kg/(m^2s)] and the interfacial area concentration A [m^2/m^3]*

$$\Gamma_i = J_i A \quad (4)$$

*The area concentration A is determined by the geometrical configuration of the gas-liquid interface and hence it is strongly dependent on the flow regime. It can be computed using simple geometrical assumptions on the dispersed phase (e.g. monodisperse bubbles with constant radius) or, for more complex flow scenarios, with additional transport equations (e.g. IATE model (Ishi and Hibiki, 2006)). The model for $J_i$ expresses the driving force for diffusive mass transfer of the component i and can be calculated with the following relationship*

$$J_i = k_i \Delta C_i \quad (5)$$

*where $k_i$ [kg/(m^2 s)] is the mass transfer coefficient, a function of the diffusion coefficient $D_i$ (Cussler, 2009; Thummala, 2016), and $\Delta C_i$ is the difference between the mass fraction of the specie in the phase ($C_i$) and at the interface ($C_{fi}$) …*